# Adaptive Curriculum Strategies: Stabilizing Reinforcement Learning for Large Language Models

## Abstract

Curriculum learning has shown promise for enhancing Large Language Models (LLMs) through progressive difficulty management, yet existing approaches suffer from instability issues when applied to reinforcement learning paradigms. Existing curriculum-based RL training exhibits catastrophic performance collapse during difficulty transitions, particularly when models encounter samples beyond their current capabilities. This instability stems from rigid curriculum designs that fail to adapt to individual model characteristics and learning trajectories. To address these limitations, we propose Adaptive Curriculum Strategies (ACS), a framework that promotes stable and effective training throughout curriculum progression. Our approach introduces model-specific difficulty calibration that adapts to each model's capabilities, and "Guided Prompting" that transforms challenging samples to prevent training instability. Experiments demonstrate that ACS prevents performance collapse in traditional curriculum RL training, achieving substantial improvements across five mathematical reasoning benchmarks while enhancing training stability.

## 1 Introduction

Large Language Models (LLMs) have achieved remarkable success in complex reasoning tasks, with reinforcement learning emerging as an effective paradigm for mathematical reasoning due to its clear reward signals and iterative optimization capabilities (Guo et al., 2024; Shao et al., 2024; Yang et al., 2025; Kavukcuoglu, 2025; OpenAI, 2024). Recent work has introduced curriculum learning strategies into RL frameworks, progressively exposing models to problems of increasing difficulty and showing promising improvements in convergence speed and performance (Wen et al., 2025b; Huang et al., 2025; Shi et al., 2025; Team et al., 2025). However, existing curriculum-based reinforcement learning approaches suffer from severe training instability, particularly during transitions between difficulty levels.

**The Challenge of Instability in Curriculum-Based Reinforcement Learning.** Current curriculum learning approaches for RL optimization face a fundamental stability problem that manifests in several ways: (1) models experience sudden performance drops during difficulty transitions, with abrupt accuracy degradation rather than smooth progression when advancing to higher difficulty levels; (2) identical curriculum strategies produce inconsistent learning trajectories across different model architectures, and some achieving stable improvement while others exhibit erratic performance fluctuations or complete learning failure under the same curriculum arrangement; and (3) models fail to maintain previously acquired capabilities when advancing to more challenging content, suffering catastrophic forgetting of simpler skills they had previously mastered.

The root cause lies in the rigid, non-adaptive nature of existing curriculum designs. As illustrated in Figure 1, difficulty perception varies dramatically across models, approximately 55% of questions that are easily solved by one model prove challenging for another. This reveals a fundamental flaw in current approaches: they rely on fixed difficulty hierarchies that assume universal difficulty perception across models (Yu et al., 2025; Wen et al., 2025a). When predefined difficulty levels are applied uniformly across diverse architectures, the mismatch between assumed and actual difficulty leads

to inappropriate sample selection, destabilizing training and explaining why identical curriculum strategies produce inconsistent results across different models.

Recent attempts to address curriculum learning in RL have focused on heuristic-based difficulty ranking approaches (Xie et al., 2025; Wen et al., 2025b), but these methods maintain the same fundamental flaw: they impose external difficulty assessments without considering the dynamic, evolving nature of individual model capabilities. This problem is particularly acute in reinforcement learning settings, where policy optimization relies on consistent reward signals and stable training trajectories.

**Our Solution: Adaptive Curriculum Strategies.** We propose Adaptive Curriculum Strategies (ACS), a framework to ensure stable curriculum-based RL training. ACS introduces two key innovations: model-specific difficulty calibration that adapts sample complexity assessment based on each model's evolving capabilities, and "Guided Prompting," a sample transformation technique that prevents catastrophic performance collapse when models encounter challenging samples.

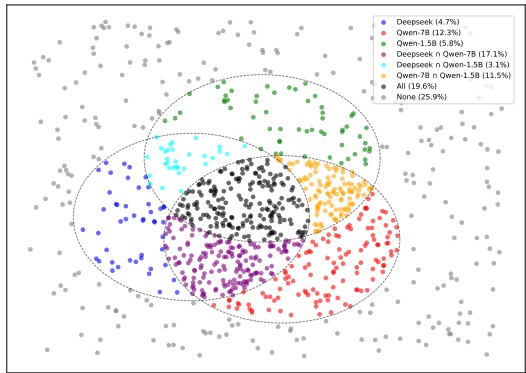

Figure 1: Solution correctness patterns for three mathematical reasoning models. Each colored region represents problems solved by specific model combinations. Our analysis shows that approximately 55% of questions that are easy for one model prove difficult for another, demonstrating that unified difficulty standards across models are problematic.

Unlike existing approaches, our framework prioritizes stability as a primary objective, ensuring progressive training throughout all curriculum stages. Our evaluation demonstrates that ACS eliminates performance instability in curriculum-based RL training while achieving superior performance across five mathematical reasoning benchmarks.

The contributions of this work include:

- We identify and characterize the training instability phenomenon in current curriculum-based RL methods, revealing that rigid unified difficulty standards cause catastrophic performance collapse due to significant difficulty perception differences across model architectures.

- We propose Adaptive Curriculum Strategies (ACS) with two key technical innovations: model-specific difficulty calibration that dynamically adapts to individual model capabilities, and guided prompting that transforms challenging samples to prevent training destabilization.

- We demonstrate through comprehensive experiments on five mathematical reasoning benchmarks that ACS eliminates training instability while achieving consistent performance improvements, validating both the stability and effectiveness of our approach.

## 2 RELATED WORK

**Curriculum Learning and Adaptive Training Strategies.** The challenge of effectively managing training sample difficulty has gained increasing attention in large language model optimization. Bengio et al. (2009) established foundational concepts for progressive difficulty management in machine learning, demonstrating that models learn more effectively when training examples are presented with appropriate difficulty sequencing. Recent work has explored various approaches to difficulty-aware training in language models. Xie et al. (2025) implements difficulty progression by adjusting task complexity based on logical reasoning requirements, while Wen et al. (2025b) identifies challenging samples based on model prediction failures and defers them to later training stages. Team et al. (2025) focuses on filtering strategies that remove problematic samples early in training, concentrating computational resources on high-quality examples. Huang et al. (2025) applies difficulty-aware training to retrieval-augmented generation, ordering tasks by the complexity of retrieved information. Shi et al. (2025) proposes dynamic sample selection based on predefined

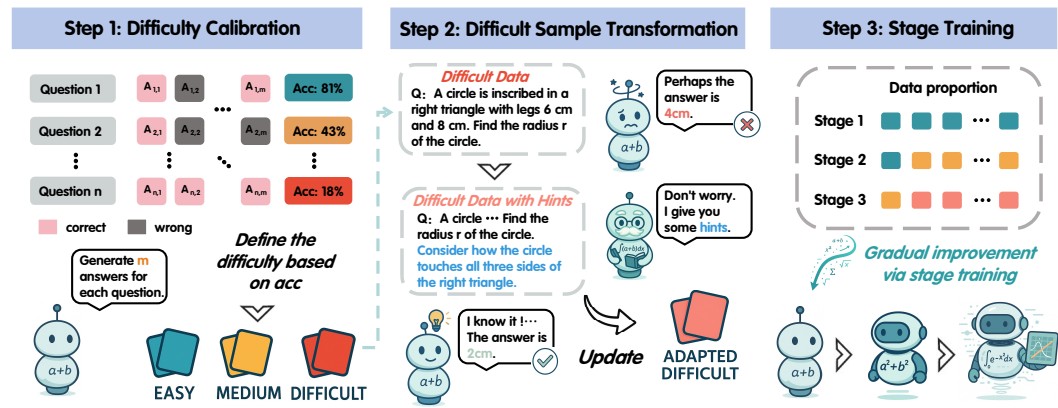

Figure 2: **Adaptive Curriculum Strategies Pipeline for Stable RL Training. Step 1:** Model-specific difficulty calibration adapts to individual model capabilities, ensuring curriculum construction aligns with each model's evolving learning capacity. **Step 2:** Guided prompting strategically transforms challenging samples to prevent training instability while preserving learning value, maintaining stable training dynamics. **Step 3:** Progressive curriculum training with data mixing strategy enables gradual improvement through staged learning and curriculum review, building capabilities from basic to advanced reasoning tasks.

difficulty scores that align with model capabilities during training. Tong et al. (2024); Xue et al. (2025) implement empirical approaches through multi-round sampling, defining difficulty through response accuracy patterns and allocating training emphasis accordingly. Ma et al. (2024) extends this concept by implementing inverse accuracy weighting, where samples with lower success rates receive proportionally greater training attention. However, these approaches rely on fixed difficulty hierarchies that fail to account for individual model capabilities and learning trajectories, often leading to training instability during curriculum transitions.

**Reinforcement Learning for Mathematical Reasoning.** Reinforcement learning has emerged as a particularly effective paradigm for mathematical reasoning tasks, where reward signals can be clearly defined through solution correctness. Luo et al. (2023); Luong et al. (2024); Yue et al. (2025) demonstrate effectiveness of Proximal Policy Optimization (PPO) for mathematical reasoning enhancement. Shao et al. (2024); DeepSeek-AI et al. (2025); Yu et al. (2025) advance the field through Group Relative Policy Optimization (GRPO), showing substantial improvements in reasoning performance. However, when integrated with curriculum learning strategies, these methods lack stability-preserving mechanisms and often suffer from catastrophic performance collapse during difficulty transitions. The effectiveness of reinforcement learning in mathematical domains makes it an ideal testbed for adaptive difficulty management strategies, as reward signals provide clear feedback on model capability progression.

## 3 METHOD

Current curriculum learning approaches in reinforcement learning suffer from a fundamental stability problem: they impose rigid difficulty progressions that fail to adapt to individual model capabilities, leading to catastrophic performance collapse during training. This instability undermines the core benefits of curriculum learning and prevents effective policy optimization in RL settings.

To address these critical limitations, we propose Adaptive Curriculum Strategies (ACS), a framework specifically designed to ensure stable, smooth, and effective curriculum-based RL training. ACS operates through three synergistic components designed to improve training stability while maintaining learning effectiveness.

Our framework is built on three fundamental principles that ensure training stability: (1) Adaptive Difficulty Assessment that calibrates sample complexity based on individual model performance rather than fixed hierarchies, (2) Stability-Preserving Sample Transformation that prevents catastrophic performance drops while preserving learning value, and (3) Progressive Adaptation that

ensures smooth transitions between curriculum stages without destabilizing training dynamics. The full algorithm is detailed in Algorithm 1.

### 3.1 Model-Specific Difficulty Calibration

The primary source of instability in curriculum-based RL training stems from the mismatch between predefined difficulty assessments and actual model capabilities. Fixed difficulty hierarchies ignore the dynamic nature of model learning and frequently expose models to inappropriate training content that destabilizes optimization.

We address this fundamental issue through model-specific difficulty calibration that adapts sample assessment to each model's evolving capabilities. Rather than relying on external difficulty labels, our approach directly measures sample accessibility through model performance. For each training sample $i$, we generate $n$ responses and evaluate their correctness against the reference answer:

$$ACC_i = \frac{\sum_{j=1}^{n} \mathbb{I}\{A_{ij} = A_i^*\}}{n} \tag{1}$$

where $A_{ij}$ represents the $j$-th generated answer for sample $i$, $A_i^*$ denotes the reference answer for sample $i$, $n$ is the number of generated responses, and $\mathbb{I}\{\}$ is the indicator function that equals 1 when the condition is true and 0 otherwise. $ACC_i$ represents the stability-informed accuracy rate for sample $i$, providing a direct measure of whether the sample is within the model's current learning capacity.

**Computational Efficiency.** While our approach requires generating multiple responses for difficulty assessment, we employ the VLLM framework for efficient inference acceleration. Taking the Qwen2.5-Math-7B model as an example, the additional computational overhead for difficulty calibration represents less than 5% of the total GPU hours compared to the entire training process, making this overhead negligible while providing substantial stability benefits.

Our approach monitors model performance across training samples, dynamically adjusting difficulty assessments as the model evolves. This adaptive mechanism ensures that the curriculum remains appropriately challenging without introducing destabilizing content that could compromise training stability.

**Calibration Robustness.** To validate the robustness of our difficulty calibration approach, we examined how variations in sampling parameters affect data partitioning consistency. Using our default configuration ($n = 16$, $T = 0.7$) as reference, we measure the overlap ratio between partitions generated with different parameters. As shown in Table 9, our results demonstrate high consistency across different parameter choices, with average overlap ratios exceeding 93% in all configurations. This indicates that our difficulty calibration produces stable and reliable sample partitioning that is robust to reasonable variations in sampling parameters.

### 3.2 Guided Prompting for Sample Transformation

**Why Existing Curriculum Approaches Fail in RL Settings.** Traditional curriculum learning faces a critical stability challenge in reinforcement learning contexts, where optimization requires consistent policy gradients and stable reward signals for effective convergence. However, traditional curriculum approaches introduce sudden difficulty transitions that violate these stability requirements, leading to catastrophic performance collapse that undermines all previous learning gains. When models encounter samples significantly beyond their capabilities, the resulting poor performance and unstable gradients can cause policy collapse and training divergence. As demonstrated in Figure 3, curriculum-based RL training without proper stability management leads to severe performance degradation. Specifically, the Qwen2.5-Math-1.5B model suffers a performance drop of approximately 30% when transitioning to Stage III, while the larger Qwen2.5-Math-7B model also exhibits notable instability, highlighting the pervasive nature of this challenge across different model scales. Our ACS framework specifically addresses these RL-specific stability requirements through adaptive curriculum management that prevents such catastrophic failures.

This catastrophic instability not only wastes computational resources but also violates the fundamental assumptions of reinforcement learning optimization, which requires stable policy gradients

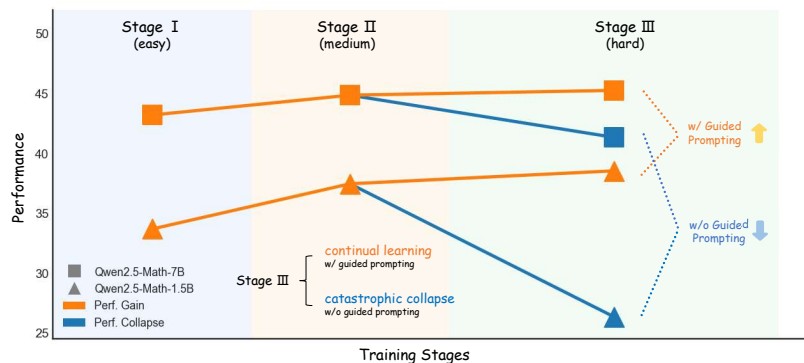

Figure 3: **Training Stability Analysis: ACS vs. Traditional Curriculum Approaches.** Our ACS framework (orange line) shows improved training progression throughout all curriculum stages, while traditional approaches (blue line) suffer significant performance degradation in Stage III. The results demonstrate the benefits of adaptive curriculum strategies for enhancing RL training stability across different model sizes: Qwen2.5-Math-1.5B (triangles) and Qwen2.5-Math-7B (squares).

for effective convergence. Our analysis reveals that models without stability-preserving mechanisms suffer dramatic performance drops that persist even after returning to easier samples, indicating fundamental damage to the learned policy.

To ensure training stability while preserving the learning value of challenging samples, we introduce "Guided Prompting," a strategic sample transformation technique designed specifically for stable RL training. Rather than discarding difficult samples or exposing models to destabilizing content, our approach transforms challenging examples into accessible learning opportunities.

For a challenging problem $Q_i$ with reference solution $S_i = \{s_{i1}, s_{i2}, ..., s_{ik}\}$, we extract a strategic guidance prefix $P_i = \{s_{i1}, s_{i2}, ..., s_{ip}\}$ where $p < k$. We gradually provide hints until either the ratio $\frac{|p|}{|k|}$ reaches a predefined hint ratio $\alpha$, or the model's performance improves to meet an accuracy threshold $\tau$. This transformation creates a stability-preserving training example:

$$y_i \sim \pi_\theta(Y|[Q_i; P_i]) \tag{2}$$

The guided prompting approach maintains training stability by ensuring that all samples remain within the model's learning capacity while preserving the educational value of challenging content. This prevents the policy instability that typically occurs when RL training encounters samples beyond model capabilities.

As shown in Figure 3, our stability-preserving approach enables consistent performance improvement throughout all curriculum stages, eliminating the catastrophic collapses that plague traditional curriculum methods and ensuring reliable RL optimization.

We provide comprehensive parameter analysis for our Guided Prompting mechanism in Appendix D.7, demonstrating the optimal balance between hint ratio and stability threshold.

## 3.3 PROGRESSIVE REINFORCEMENT LEARNING

Using our stability-informed difficulty calibration and sample transformation, we implement progressive reinforcement learning that maintains training stability throughout curriculum progression. The training data is adaptively partitioned into stability-ordered subsets $D = \{D_1, D_2, ..., D_p\}$ where each subset $D_j$ contains samples verified to be accessible at the current model capability level.

Reinforcement learning provides particularly effective optimization for mathematical reasoning tasks due to clear reward signals from solution correctness. Following successful approaches in mathematical reasoning DeepSeek-AI et al. (2025), we implement staged reinforcement learning that can operate directly on pretrained models.

---

**Algorithm 1** Adaptive Curriculum Strategies for Stable RL Training

---

**Require:** training dataset $D = \{(Q_i, A_i)\}$ with questions $Q_i$ and reference solution $A_i$
**Require:** pretrained model $\pi_0$, stability threshold $\tau$, adaptation ratio $\alpha$
1: **Adaptive Curriculum Construction with Stability Monitoring:**
2: **for all** question $Q_i \in D$ **do**
3:      Generate $n$ responses $\{A_{i1}, \ldots, A_{in}\}$ using model $\pi_\theta$
4:      Calculate stability-informed accuracy: $ACC_i = \frac{\sum_{j=1}^n \mathbb{I}\{A_{ij}=A_i^*\}}{n}$
5:      Monitor performance variance to detect potential instability indicators
6: **end for**
7: Adaptively partition dataset $D$ into stability-ordered subsets $\{D_1, D_2, ..., D_p\}$ based on model-specific accessibility
8: **Stability-Preserving Sample Transformation:**
9: **for all** challenging samples $(Q_i, A_i) \in D_p$ **do**
10:      Decompose solution $S_i$ into guided steps $\{s_{i1}, \ldots, s_{ik}\}$
11:      Apply guided prompting: gradually provide hints $P_i = \{s_{i1}, \ldots, s_{il}\}$
12:      Monitor stability: continue until performance reaches $\tau$ or adaptation ratio reaches $\alpha$
13:      **if** stability threshold achieved **then**
14:          Transform sample: $Q_i \rightarrow [Q_i; P_i]$, $A_i \rightarrow \{s_{i(l+1)}, \ldots, s_{ik}\}$
15:      **else**
16:          Defer sample to later stage with additional adaptation
17:      **end if**
18: **end for**
19: **Stable Progressive RL Training:**
20: **for** each curriculum stage $s \in \{1, \ldots, p\}$ **do**
21:      Apply RL optimization on stability-adapted dataset $D_s$:

$$\pi_s = \arg \min_{(Q,A) \in D_s} \mathcal{L}_{RL}(\pi_{s-1}) \text{ with stability constraints}$$

22: **end for**
23: **Output:** Stably trained model $\pi_m$ with enhanced reasoning capabilities

---

Our reinforcement learning implementation incorporates stability monitoring at every stage to prevent the catastrophic collapses observed in traditional curriculum approaches. We employ Group Relative Policy Optimization (GRPO) with additional stability constraints that ensure consistent policy improvement without destabilizing training dynamics.

The reward function maintains both accuracy and formatting components while incorporating stability considerations:

$$r_{format} = \begin{cases} 1.0 & \text{if format is correct} \\ 0.0 & \text{otherwise} \end{cases} \tag{3}$$

$$r_{accuracy} = \begin{cases} 1.0 & \text{if prediction is correct} \\ 0.0 & \text{otherwise} \end{cases} \tag{4}$$

The total reward $r = r_{format} + r_{accuracy}$ provides stable optimization signals that enable consistent policy improvement without the instability issues that plague traditional curriculum-based RL training.

Our stability-aware GRPO implementation generates multiple candidate responses $O = \{o_1, o_2, ..., o_G\}$ for each question while monitoring training stability. The relative advantages are computed as:

$$A_i = \frac{r_i - mean(\{r_1, r_2, ..., r_G\})}{std(\{r_1, r_2, ..., r_G\})} \tag{5}$$

The GRPO objective incorporates clipping and KL regularization with additional stability constraints:

$$\mathcal{L}_{GRPO} = \mathbb{E}_{(x) \sim D} \left[ \frac{1}{G} \sum_{i=1}^{G} \frac{1}{|o_i|} \sum_{t=1}^{|o_i|} (min(r_{i,t}(\theta) A_i,\right.$$
$$\left. clip(r_{i,t}(\theta), 1 - \epsilon, 1 + \epsilon) A_i) - \beta D_{KL}(\pi_\theta || \pi_{ref})) \right], \tag{6}$$

where $r_{i,t}(\theta) = \frac{\pi_\theta(o_{i,t}|q,o_{i,<t})}{\pi_{\theta_{old}}(o_{i,t}|q,o_{i,<t})}$ represents the importance sampling ratio.

This approach ensures that the RL optimization remains stable and effective throughout all curriculum stages, eliminating the performance collapses that have hindered previous curriculum-based approaches and enabling reliable policy optimization in challenging domains.

## 4 EXPERIMENTS

### 4.1 EXPERIMENTAL SETUP

We evaluate our ACS framework using mathematical reasoning tasks, testing on two model scales with GRPO across five benchmark datasets. Complete experimental details including dataset construction, model configurations, training hyperparameters, and evaluation protocols are provided in the Appendix A.

### 4.2 COMPARATIVE METHODS AND EXPERIMENTAL DESIGN

To systematically evaluate our ACS framework, we design comprehensive comparisons across two key dimensions of curriculum learning.

#### 4.2.1 DIFFICULTY CALIBRATION STRATEGIES

We compare our model-specific difficulty calibration against three established approaches:

**Light-R1.** Following Wen et al. (2025b), this method employs a fixed external model to assess sample difficulty. Difficulty rankings are determined based on the external model's success rates, representing a model-agnostic assessment strategy that does not account for target model capabilities.

**Length-based Ranking.** This heuristic approach ranks samples based on solution length, assuming longer solutions correspond to more complex problems. Samples are organized in ascending order of reference solution lengths.

**Original Dataset Labels.** This approach utilizes the predefined difficulty levels (1-5) provided by the MATH dataset, organizing training samples according to expert-defined difficulty hierarchies without considering individual model performance.

#### 4.2.2 CHALLENGING SAMPLE PROCESSING STRATEGIES

For samples exceeding model capabilities, we evaluate two alternative processing approaches:

**Retain All Samples.** This approach includes all difficult samples without modification, potentially exposing models to training instability from inaccessible examples.

**Discard Difficult Samples.** This conservative strategy removes samples below a predefined accuracy threshold, avoiding negative impacts but discarding valuable training data.

### 4.3 MAIN RESULTS

As shown in Table 1, the ACS framework exhibits both methodological robustness and cross-model scalability. Compared to the GRPO baseline, the improvements were pronounced at $55.9\%$ (from 24.7 to 38.5) and $5.8\%$ (from 42.8 to 45.3) for Qwen2.5-Math-1.5B and Qwen2.5-Math-7B respectively, demonstrating the particular effectiveness of adaptive difficulty management in reinforcement

Table 1: Main experimental results comparing ACS with baseline methods and ablation studies across two model scales. Results show performance on five mathematical reasoning benchmarks using GRPO training. ACS demonstrates consistent improvements through model-specific difficulty calibration and guided prompting for challenging samples.

| Model | Method | MATH 500 | Minerva Math | Olympiad Bench | AIME24 | AMC23 | Average |
|---|---|---|---|---|---|---|---|
| Qwen2.5 Math 1.5B | Base Model | 42.8 | 7.7 | 25.2 | 3.3 | 22.5 | 20.3 |
| | GRPO | 51.8 | 18.4 | 21.0 | 10.0 | 22.5 | 24.7 |
| | *Difficulty Calibration Methods* | | | | | | |
| | Light-R1 | 64.2 | 31.2 | 30.5 | 6.7 | 40.0 | 34.5 |
| | Length | 67.4 | **33.8** | 28.7 | 6.7 | 32.5 | 33.8 |
| | Original | 57.8 | 23.9 | 23.7 | 6.7 | 27.5 | 27.9 |
| | *Challenging Sample Processing* | | | | | | |
| | Retain | 62.2 | 17.3 | 26.5 | 3.3 | 22.5 | 26.4 |
| | Discard | 72.0 | 30.9 | 30.1 | 10.0 | 40.0 | 36.6 |
| | *ACS (Ours)* | **72.6** | 31.6 | **32.7** | **13.3** | **42.5** | **38.5** |
| Qwen2.5 Math 7B | Base Model | 63.6 | 12.5 | 25.8 | 10.0 | 42.5 | 30.9 |
| | GRPO | 74.2 | 33.5 | 33.9 | 10.0 | **62.5** | 42.8 |
| | *Difficulty Calibration Methods* | | | | | | |
| | Light-R1 | 73.2 | **43.0** | 37.8 | **13.3** | 55.0 | 44.5 |
| | Length | 75.2 | 40.8 | 34.7 | **13.3** | 57.5 | 44.3 |
| | Original | 75.6 | 35.7 | 35.7 | **13.3** | 45.0 | 41.1 |
| | *Challenging Sample Processing* | | | | | | |
| | Retain | 71.8 | 41.5 | 35.9 | 10.0 | 47.5 | 41.3 |
| | Discard | **77.4** | 37.5 | 37.3 | **13.3** | 55.0 | 44.1 |
| | *ACS (Ours)* | 76.6 | 38.2 | **38.2** | **13.3** | 60.0 | **45.3** |

learning paradigms. Furthermore, ACS achieves the highest average performance across both model scales, demonstrating superior overall effectiveness compared to other baseline methods, including difficulty calibration strategies (Light-R1, Length-based, Original) and challenging sample processing approaches (Retain, Discard). Notably, our ACS training strategy yielded generally consistent performance gains across all evaluation benchmarks, confirming its effectiveness in enhancing model generalization through strategic difficulty adaptation.

## 4.4 ABLATION STUDIES

### 4.4.1 CROSS-MODEL GENERALIZABILITY

To address concerns about the generalizability of our approach across different model architectures, we conducted additional experiments using DeepSeek-Math-7B-Instruct as the base model. This evaluation is particularly important given that prior work has shown some models may exhibit varying sensitivity to training signals in reinforcement learning paradigms. Additionally, to provide comprehensive comparison baselines, we include results from Light-R1 (the best performing difficulty calibration method from our main experiments) and Discard (the best performing challenging sample processing strategy from our comparative analysis).

As demonstrated in Table 2, our ACS framework maintains its effectiveness when applied to the DeepSeek-Math model architecture and achieves the highest average performance across all methods. Compared to the GRPO baseline, ACS achieves a significant improvement of 8.9% (from

17.9 to 19.5). ACS also outperforms other competitive methods, with improvements of 5.4% over Light-R1 (from 18.5 to 19.5) and 4.3% over the Discard strategy (from 18.7 to 19.5).

Table 2: Cross-model evaluation results on DeepSeek-Math-7B-Instruct model, comparing ACS with the best-performing baseline methods from each category

| Model | Method | MATH 500 | Minerva Math | Olympiad Bench | AIME24 | AMC23 | Average |
|---|---|---|---|---|---|---|---|
| **DeepSeek Math 7B Instruct** | *GRPO* | 39.6 | 18.0 | 13.6 | **3.3** | 15.0 | 17.9 |
| | *Light-R1* | 40.2 | 20.6 | 14.0 | 0.0 | **17.5** | 18.5 |
| | *Discard* | 40.8 | 21.4 | 13.2 | **3.3** | 15.0 | 18.7 |
| | *ACS(Ours)* | **41.4** | **22.8** | **14.8** | **3.3** | 15.0 | **19.5** |

These results validate that our proposed ACS framework exhibits robust cross-model generalizability, addressing the limitation of model-specific optimization strategies. The consistent performance improvements across different architectural foundations demonstrate that the core principles of model-adaptive curriculum construction and guided prompting are broadly applicable to various mathematical reasoning models, rather than being artifacts of specific model characteristics.

### 4.4.2 ALGORITHM GENERALIZABILITY

To validate the generalizability of ACS beyond GRPO, we conducted additional experiments using Proximal Policy Optimization (PPO) on the Qwen2.5-Math-1.5B model. PPO represents the most classic and widely-adopted RL baseline in the field. As shown in Table 3, ACS demonstrates consistent stability-preserving benefits across different RL algorithms, with PPO w/ ACS achieving 25.5% improvement (from 28.2 to 35.4) over vanilla PPO.

Table 3: Generalizability evaluation using PPO algorithm on Qwen2.5-Math-1.5B model

| Method | MATH 500 | Minerva Math | Olympiad Bench | AIME24 | AMC23 | Average |
|---|---|---|---|---|---|---|
| PPO | 65.6 | 16.8 | 19.5 | 6.7 | 32.5 | 28.2 |
| PPO w/ ACS | **69.4** | **28.7** | **28.9** | **10.0** | **40.0** | **35.4** |

### 4.4.3 IMPACT OF GUIDED PROMPTING

To validate the effectiveness of our Guided Prompting mechanism, we conducted direct comparison experiments between models trained with and without this component. A key concern is whether hint-augmented training might create dependency on external guidance during inference.

As shown in Table 4, all test sets contain no hints. The results demonstrate that guided prompting significantly improves model performance across all benchmarks, with the Qwen2.5-Math-1.5B model achieving 45.8% improvement (from 26.4 to 38.5). This validates that our stability-preserving sample transformation enables models to internalize effective problem-solving strategies from challenging content without creating inference-time dependency.

### 4.4.4 DATA MIXING STRATEGY

Drawing inspiration from human learning processes where students periodically review previously mastered knowledge, we investigate whether models undergoing staged curriculum learning require similar reinforcement of previously acquired content. This analysis is particularly crucial for maintaining training stability throughout the curriculum progression. We design two distinct data mixing strategies for comparative analysis within our ACS framework:

Table 4: Impact of Guided Prompting on model performance. All test sets contain NO hints, providing direct assessment of independent problem-solving ability. "w/o Guided Prompting" corresponds to the "Retain" baseline that includes all difficult samples without modification.

| Model | Training Strategy | MATH 500 | Minerva Math | Olympiad Bench | AIME24 | AMC23 | Average |
|-------|-------------------|----------|--------------|----------------|--------|-------|---------|
| Qwen2.5 Math 1.5B | w/o Guided Prompting | 62.2 | 17.3 | 26.5 | 3.3 | 22.5 | 26.4 |
| | w/ Guided Prompting | **72.6** | **31.6** | **32.7** | **13.3** | **42.5** | **38.5** |
| Qwen2.5 Math 7B | w/o Guided Prompting | 71.8 | 41.5 | 35.9 | 10.0 | 47.5 | 41.3 |
| | w/ Guided Prompting | **76.6** | **38.2** | **38.2** | **13.3** | **60.0** | **45.3** |

**Naive Curriculum:** Models receive samples corresponding only to the current difficulty level at each training stage, focusing exclusively on new, challenging content without revisiting previously learned material.

**Curriculum Review:** A strategic data mixing approach that incorporates a small proportion of easier samples from previous stages during later training phases, allowing the model to revisit and reinforce previously acquired capabilities while learning new content.

Table 5: Data mixing strategy comparison: Naive Curriculum vs. Curriculum Review across different model scales. Bold numbers indicate superior performance. Curriculum Review incorporates 10% of samples from the previous difficulty level at each training stage.

| Model | Method | MATH 500 | Minerva Math | Olympiad Bench | AIME24 | AMC23 | Average |
|-------|--------|----------|--------------|----------------|--------|-------|---------|
| Qwen2.5 Math 1.5B | *Naive Curriculum* | 69.8 | 33.8 | 30.8 | 6.7 | 22.5 | 32.7 |
| | *Curriculum Review* | **72.6** | 31.6 | **32.7** | **13.3** | **42.5** | **38.5** |
| Qwen2.5 Math 7B | *Naive Curriculum* | 75.2 | 35.7 | 36.4 | 13.3 | 52.5 | 42.6 |
| | *Curriculum Review* | **76.6** | **38.2** | **38.2** | 13.3 | **60.0** | **45.3** |

Experimental results in Table 5 demonstrate that the Curriculum Review strategy consistently outperforms the Naive Curriculum approach and achieves the best performance across both model scales. For the 1.5B model, Curriculum Review achieves a significant 17.7% average performance improvement compared to Naive Curriculum (from 32.7 to 38.5), while the 7B model shows a 6.1% improvement (from 42.6 to 45.2). These results confirm that incorporating previously learned content during later training stages prevents catastrophic forgetting and maintains training stability, aligning with our ACS stability principles.

### 4.4.5 ADDITIONAL ABLATION STUDIES

We conducted additional ablation studies to validate other design choices: (1) comparison with dynamic curriculum baseline AdaRFT (Table 6), (2) curriculum stage granularity analysis (Table 7), and (3) difficulty stability across training stages (Table 8). Complete results are provided in Appendix D.

## 5 CONCLUSIONS

We presented Adaptive Curriculum Strategies (ACS), a framework that addresses critical instability issues in curriculum-based reinforcement learning through model-specific difficulty calibration and guided prompting techniques. Our experimental results demonstrate that ACS maintains training stability while achieving superior performance across five mathematical reasoning benchmarks.

This work establishes training stability as a fundamental requirement for curriculum learning, opening promising avenues for developing more reliable training methodologies across challenging domains where progressive learning strategies are crucial for optimal performance.

ETHICS STATEMENT

This research focuses on mathematical reasoning tasks and adheres to the ICLR Code of Ethics. Our work addresses a technical training stability problem without involving human subjects, personal data, or sensitive applications. All datasets are publicly available academic benchmarks. The research process follows established ethical guidelines and poses no ethical concerns.

REPRODUCIBILITY STATEMENT

Our work ensures strong reproducibility through comprehensive documentation and open-source resources. All models (Qwen2.5-Math series, DeepSeek-Math), datasets (MATH, OlympiadBench, etc.), and training frameworks (Hugging Face Open R1, VLLM) are publicly available. Complete training hyperparameters are detailed in Appendix A.4, data construction procedures in Appendix A.1, and experimental procedures throughout the appendix. The straightforward implementation using standard frameworks makes reproduction simple and accessible.

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

## A  EXPERIMENTAL DETAILS

### A.1  DATASET CONSTRUCTION

Following the experimental setting of Zeng et al. (2025), we selected the MATH dataset Hendrycks et al. (2021) and extracted samples from level 3 to level 5 as training data, comprising a total of 9,255 instances. To implement our proposed ACS framework for creating model-specific difficulty calibration, we need to feed all training set samples into the pre-trained model for inference and evaluate the model's accuracy on each sample.

To ensure that the evaluation results are as reliable as possible while not causing excessive computational overhead, for each question in the dataset, we use the VLLM framework to generate 16 responses from the model, extract predictions from these responses using appropriate scripts, and compare them with golden answers to determine the correctness of the generations. To fully harness the model's potential, we did not adopt a greedy decoding strategy to generate responses, but instead set the temperature to 0.7, generating responses through sampling.

After calculating the model's accuracy on the samples through the above steps, we sort the samples and divide them into 3 equal parts according to quantity. The top 1/3 with the highest accuracy are classified as accessible samples, used for the first stage of model training. The bottom 1/3 with the lowest accuracy are classified as challenging samples, used for the final stage of model training.

In addition, for particularly challenging samples, we employed our "Guided Prompting" approach to reduce the difficulty for the model while preserving learning value. Specifically, we first collected reference answers for these challenging samples, then segmented these reference answers into step-by-step reasoning processes, as illustrated in Figure 4. Finally, we selected a small portion of the prefix combined with the original question as input to assist the model in solving problems more effectively while maintaining training stability.

All data was processed into a conversational format.

### A.2  MODEL SELECTION AND HARDWARE SETUP

To effectively validate the efficacy of our ACS method across foundation models of varying capabilities, we selected three different models for our experiments: Qwen2.5-MATH-1.5B Yang et al. (2024), Qwen2.5-MATH-7B, and DeepSeek-Math-7B-Instruct for cross-model generalizability validation. We conducted our experiments using 8 NVIDIA A100 GPUs for the GRPO experiments within Hugging Face's Open R1 framework Face (2025).

### A.3  COMPUTATIONAL OVERHEAD ANALYSIS

Our ACS framework requires evaluating model performance on training samples through multiple sampling, which introduces additional computational overhead. To minimize this cost while maintaining evaluation reliability, we employed the VLLM framework Kwon et al. (2023) for efficient model inference acceleration.

Thanks to VLLM's optimized memory management and dynamic batching capabilities, the additional time overhead introduced by our curriculum construction is minimal. Taking the Qwen2.5-Math-7B model as an example, the sample evaluation phase for difficulty calibration requires less than 5% of the total GPU hours compared to the entire training process on NVIDIA A100 GPUs, making the overhead negligible compared to the overall computational cost.

Specifically, the time cost breakdown is as follows:

- **Sample evaluation phase**: Using VLLM, we generate 16 responses per sample for the 9,255 training instances, totaling approximately 148,080 inference calls.
- **Training phase**: Standard fine-tuning process using GRPO on the curriculum-organized data.

This efficient implementation ensures that the benefits of model-specific difficulty calibration can be achieved without significant computational burden, making our approach practical for real-world applications.

## A.4 TRAINING HYPERPARAMETERS

**GRPO Training Details.** We conducted our experiments using bf16 precision under the Deep-Speed framework with zero-2 configuration. We set per_device_train_batch_size to 16 and gradient_accumulation_steps to 8, employing a cosine lr_scheduler with warmup set to 0.1 and beta to 0.04, num_generations to 7, max_prompt_length to 512 and max_completion_length 1024. For the Qwen2.5-Math-1.5B model, we used a learning rate of 3e-6 and trained for 6 epochs. For the Qwen2.5-Math-7B model, we used a learning rate of 3e-6 and trained for 4 epochs. For the DeepSeek-Math-7B-Instruct model, we used the same configuration as the Qwen2.5-Math-7B model (learning rate of 3e-6, 4 epochs).

## A.5 EVALUATION

We evaluated our models using the evaluation script from Zeng et al. (2025). For evaluation, we use five benchmark datasets to assess the model's performance across different levels of difficulty and mathematical reasoning:

- **MATH 500** Lightman et al. (2023): A subset of the MATH dataset, containing 500 representative problems designed to test a model's general mathematical capability.
- **OlympiadBench** He et al. (2024): Includes a collection of problems from Olympiad-level mathematics and physics competitions.
- **Minerva Math** Lewkowycz et al. (2022): A curated set of undergraduate-level math problems that assess complex mathematical reasoning and symbolic manipulation.
- **AMC 23** and **AIME 24**: Include problems from the 2023 American Mathematics Competitions and the 2024 American Invitational Mathematics Examination, respectively.

> **Case Study**
>
> **Problem**
> Let n be the smallest positive integer that satisfies the following conditions:
> n divided by 2 is a perfect square
> n divided by 3 is a perfect cube
> n divided by 5 is a perfect fifth power
> How many divisors of n are NOT multiples of 10?
>
> **Solution**
> The first condition implies that the power of each prime factor of $n$ must be an even power (excluding $2$, which must be an odd power). The second condition implies that the power of each prime factor of $n$ must be divisible by $3$ (excluding $3$, which must leave a residue of $1$ upon division by $3$). The third condition implies that the power of each prime factor of $n$ must be divisible by $5$ (excluding $5$, which must leave a residue of $1$ upon division by $5$).\nClearly, to minimize $n$, we want to just use the prime factors $2,3,5$. The power of $2$ must be divisible by $3,5$, and $2^{15}$ works. Similarly, the powers of $3$ and $5$ must be $10$ and $6$, respectively, both of which leave a residue of $1$ upon division. Thus, we need the number of factors of $2^{15} \\cdot 3^{10} \\cdot 5^{6}$ which are not multiples of $10$.\nApplying the complement principle, there are a total of $(15+1)(10+1)(6+1) = 1232$ factors. We can draw a bijection between the number of divisors of $2^{15} \\cdot 3^{10} \\cdot 5^{6}$ that are divisible by $10$ and the number of divisors of $2^{14} \\cdot 3^{10} \\cdot 5^{5}$ (as each of these divisors, when multiplied by 10, will provide a factor of the original number that is divisible by 10). There are $(14+1)(10+1)(5+1) = 990$. The answer is $1232-990 = \\boxed{242}$.
>
> **Step 1**
> The first condition implies that the power of each prime factor of $n$ must be an even power (excluding $2$, which must be an odd power).
> **Step 2**
> The second condition implies that the power of each prime factor of $n$ must be divisible by $3$ (excluding $3$, which must leave a residue of $1$ upon division by $3$).
> **Step 3**
> The third condition implies that the power of each prime factor of $n$ must be divisible by $5$ (excluding $5$, which must leave a residue of $1$ upon division by $5$).
> **Step 4**
> Clearly, to minimize $n$, we want to just use the prime factors $2,3,5$.
> **Step 5**
> The power of $2$ must be divisible by $3,5$, and $2^{15}$ works.
> **Step 6**
> Similarly, the powers of $3$ and $5$ must be $10$ and $6$, respectively, both of which leave a residue of $1$ upon division.
> **Step 7**
> Thus, we need the number of factors of $2^{15} \cdot 3^{10} \cdot 5^{6}$ which are not multiples of $10$.
> **Step 8**
> Applying the complement principle, there are a total of $(15+1)(10+1)(6+1) = 1232$ factors.
> **Step 9**
> We can draw a bijection between the number of divisors of $2^{15} \cdot 3^{10} \cdot 5^{6}$ that are divisible by $10$ and the number of divisors of $2^{14} \cdot 3^{10} \cdot 5^{5}$ (as each of these divisors, when multiplied by 10, will provide a factor of the original number that is divisible by 10).
> **Step 10**
> There are $(14+1)(10+1)(5+1) = 990$.
> **Step 11**
> The answer is $1232-990 = \boxed{242}$.

Figure 4: Decomposition of reference answers into step-by-step solution.

## B PROMPT DETAILS

During both training and testing processes, the data was processed into a conversational format. Figure 5 demonstrate the prompts we used during the GRPO processes respectively. After training the models using their respective methods, we employed the corresponding prompts during testing as well. Additionally, during the GRPO training process, besides adding the User's description, we also appended part of the Assistant's content prefixed with the special token <think>. This approach

```
GRPO Prompt

You are a helpful AI Assistant that provides well–reasoned and detailed responses.
You first think about the reasoning process as an internal monologue and then
provide the user with the answer.
Respond in the following format:
<think>
reasoning process here
</think>
<answer>
answer here
</answer>
User:
{Problem}
Assistant:
Let me solve this step by step.
<think>
```

Figure 5: Prompt template used in GRPO training for ACS implementation.

helps the model quickly learn format compliance during the reinforcement learning process, greatly enhancing the stability of the model's reinforcement learning training within our ACS framework.

## C  USE OF LARGE LANGUAGE MODELS

This paper did not use Large Language Models for writing assistance or content generation.

## D  ADDITIONAL RESULTS

### D.1  LIMITATIONS OF PREDEFINED DIFFICULTY METRICS.

Predefined difficulty metrics exhibit fundamental flaws that undermine their effectiveness in curriculum learning applications. First, these metrics lack precision in capturing actual problem complexity as experienced by language models. As demonstrated in Figure 6, our systematic evaluation on the MATH dataset reveals that model performance does not correlate with predefined difficulty rankings. Most notably, models consistently achieve higher accuracy on supposedly more difficult Level 5 problems compared to Level 4 problems, directly contradicting the assumed difficulty hierarchy. This counterintuitive pattern indicates that expert-defined difficulty levels may not align with the computational challenges actually faced by neural models.

Second, the assumption of universal difficulty standards proves fundamentally flawed in practice. Our analysis reveals significant variation in how different models perceive problem complexity, with difficulty assessments that effectively characterize challenge levels for one architecture often failing to generalize to other models. This model-specific variation in difficulty perception explains why curriculum strategies based on fixed difficulty rankings produce inconsistent training outcomes across different architectures, highlighting the critical need for adaptive, model-aware difficulty calibration approaches.

### D.2  TRAINING PROGRESSION

In this section, we demonstrate the overall performance changes on the test set when applying our ACS framework to Qwen2.5-Math-1.5B and Qwen2.5-Math-7B using reinforcement learning methods for multi-stage training. As shown in Figure 7, our ACS method maintains stable performance progression as training iterations advance, demonstrating the effectiveness of our stability-preserving curriculum design.

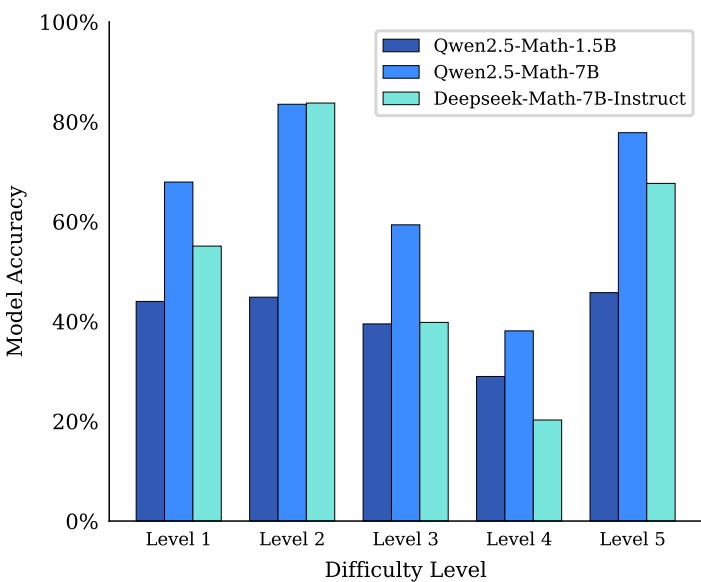

Figure 6: Performance of multiple models on MATH dataset subsets with predefined difficulty levels. As predefined difficulty increases from Level 1 to Level 5, model accuracy does not consistently decline but instead exhibits significant fluctuations, demonstrating that predefined difficulty standards may not correctly adapt to all models.

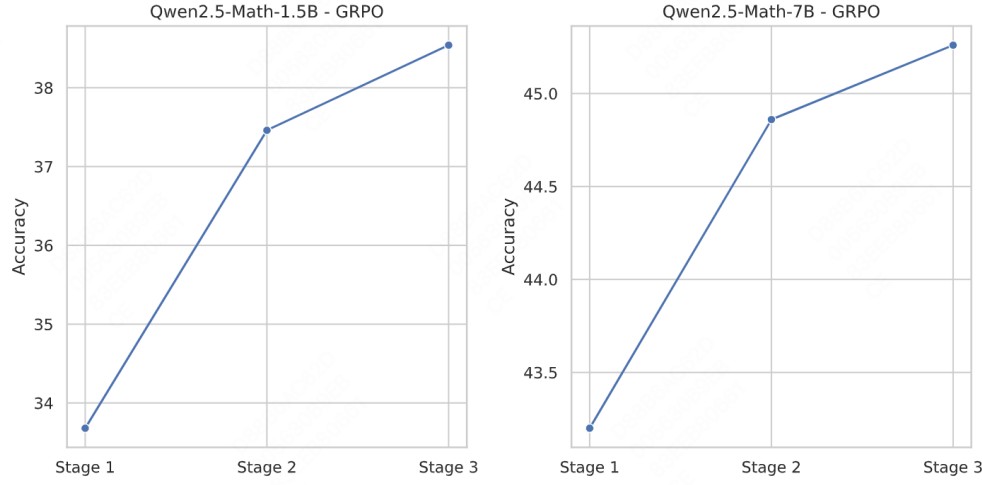

Figure 7: Performance progression across training stages using ACS framework, demonstrating stable improvement without catastrophic collapse.

### D.3 DYNAMIC CURRICULUM LEARNING COMPARISON

To provide comprehensive comparison with state-of-the-art adaptive curriculum approaches, we evaluated AdaRFT Shi et al. (2025), which implements dynamic curriculum learning that continuously adapts training data to the model's evolving capabilities. As shown in Table 6, while AdaRFT demonstrates effectiveness through dynamic scheduling, our ACS framework achieves superior performance through the combination of model-specific difficulty calibration and guided prompting mechanisms. Compared with AdaRFT, ACS achieves improvements of 3.8% (from 37.1 to 38.5) for Qwen2.5-Math-1.5B. These results demonstrate that our sample transformation mechanism provides complementary benefits beyond pure dynamic scheduling strategies, validating the effective-

Table 6: Comparison with AdaRFT dynamic curriculum baseline across two model scales

| Model | Method | MATH 500 | Minerva Math | Olympiad Bench | AIME24 | AMC23 | Average |
|-------|--------|----------|--------------|----------------|--------|-------|---------|
| Qwen2.5 Math 1.5B | AdaRFT | 68.8 | 29.4 | 32.1 | 10.0 | **45.0** | 37.1 |
| | ACS (Ours) | **72.6** | **31.6** | **32.7** | **13.3** | 42.5 | **38.5** |

ness of combining adaptive difficulty assessment with stability-preserving sample transformation.

### D.4 NUMBER OF CURRICULUM STAGES

We conducted ablation experiments on the number of curriculum stages to examine the impact of curriculum granularity. As shown in Table 7, the 3-stage configuration achieves optimal performance (38.5 average), while both 2-stage (33.9) and 4-stage (36.8) configurations show degraded results. Too few stages (2) provide insufficient progressive adaptation, while too many stages (4) may introduce unnecessary complexity and transition overhead.

Table 7: Ablation study on the number of curriculum stages using Qwen2.5-Math-1.5B model

| Number of Stages | MATH 500 | Minerva Math | Olympiad Bench | AIME24 | AMC23 | Average |
|------------------|----------|--------------|----------------|--------|-------|---------|
| 2 Stages | 67.6 | 27.6 | 30.1 | 6.7 | 37.5 | 33.9 |
| 3 Stages | **72.6** | **31.6** | **32.7** | **13.3** | **42.5** | **38.5** |
| 4 Stages | 71.8 | 30.8 | 31.5 | 10.0 | 40.0 | 36.8 |

### D.5 DIFFICULTY STABILITY ACROSS TRAINING STAGES

To address concerns about the dynamic nature of difficulty assessment, we conducted an additional experiment to track how sample difficulty changes across training stages. Specifically, we measured the overlap of samples within each difficulty level before and after each training stage. As shown in Table 8, the results demonstrate that after multiple training stages, the majority of samples remain within their original difficulty levels, with average overlap ratios exceeding 80% across all stages.

Table 8: Difficulty level stability across training stages. The table shows the percentage of samples that remain in the same difficulty level after each training stage, demonstrating the robustness of our difficulty calibration.

| Training Stage Average | Level 1 Overlap | Level 2 Overlap | Level 3 Overlap |
|------------------------|-----------------|-----------------|-----------------|
| Initial | 100% | 100% | 100% |
| After Stage 1 | 87.4% | 75.7% | 91.6% |
| After Stage 2 | 82.7% | 80.5% | 88.4% |
| After Stage 3 | 81.2% | 79.6% | 84.1% |

Further analysis reveals that difficulty transitions occur predominantly between adjacent levels. From initial to Stage 3, only 1.3% of Level 1 samples migrated to Level 3. Notably, difficult samples (Level 3) exhibit the highest stability, with 84.1% remaining in Level 3 after all training stages. This

observation reinforces the importance of proper adaptation strategies for difficult samples through our Guided Prompting mechanism.

### D.6 CALIBRATION ROBUSTNESS ANALYSIS

We conducted comprehensive robustness analysis to examine how variations in sampling parameters affect the consistency of our difficulty calibration. Using our default configuration ($n = 16$, $T = 0.7$) as reference, we measure the overlap ratio between partitions generated with different parameters.

Table 9: Calibration robustness to sampling parameters. Overlap ratios show the percentage of samples that remain in the same difficulty level when using different sampling configurations.

| Configuration | Stage I Overlap | Stage II Overlap | Stage III Overlap | Average Overlap |
|---|---|---|---|---|
| *Robustness to Number of Samples (n)* | | | | |
| $n = 8$ | 94.2% | 91.5% | 93.8% | 93.2% |
| $n = 16$ (ours) | 100% | 100% | 100% | 100% |
| $n = 32$ | 97.5% | 95.1% | 96.9% | 96.5% |
| *Robustness to Sampling Temperature (T)* | | | | |
| $T = 0.5$ | 95.3% | 92.8% | 94.6% | 94.2% |
| $T = 0.7$ (ours) | 100% | 100% | 100% | 100% |
| $T = 0.9$ | 96.1% | 93.5% | 95.2% | 94.9% |

As demonstrated in Table 9, our results show high consistency across different parameter choices, with average overlap ratios exceeding 93% in all configurations. This indicates that our difficulty calibration approach produces stable and reliable sample partitioning that is robust to reasonable variations in sampling parameters. The configuration ($n = 16$, $T = 0.7$) achieves an optimal balance between reliable assessment and computational efficiency while maintaining stability.

Table 10: Ablation study on Guided Prompting parameters using Qwen2.5-Math-1.5B model. The results demonstrate the optimal balance achieved by our default configuration ($\alpha = 25\%$, $\tau = 20\%$).

| Configuration | MATH 500 | Minerva Math | Olympiad Bench | AIME24 | AMC23 | Average |
|---|---|---|---|---|---|---|
| $\alpha = 10\%, \tau = 20\%$ | 71.2 | 30.2 | 31.8 | 10.0 | 40.0 | 36.6 |
| $\alpha = 25\%, \tau = 20\%$ (ours) | **72.6** | **31.6** | **32.7** | **13.3** | **42.5** | **38.5** |
| $\alpha = 40\%, \tau = 20\%$ | 69.4 | 29.8 | 29.4 | 10.0 | 35.0 | 34.7 |
| $\alpha = 25\%, \tau = 30\%$ | 70.8 | 30.5 | 30.9 | 13.3 | 40.0 | 37.1 |

### D.7 GUIDED PROMPTING PARAMETER ANALYSIS

We conducted comprehensive ablations over hint ratio ($\alpha$) and accuracy threshold ($\tau$) to analyze the trade-offs in our Guided Prompting mechanism. In all our main experiments, we set $\alpha = 25\%$ and $\tau = 20\%$.

As shown in Table 10, the results reveal a clear trade-off in hint ratio selection: (1) Too few hints ($\alpha = 10\%$) under-utilize the stabilization benefit, limiting the model's ability to learn from challenging problems; (2) Too many hints ($\alpha = 40\%$) lead to performance degradation, suggesting distribution shift where models become overly dependent on hint structure; (3) The optimal balance at $\alpha = 25\%$ provides sufficient scaffolding while maintaining adequate challenge. During inference, our models receive only original problem statements without any hints, and the strong generalization results validate that models do not develop inference-time dependency on hint structures.

