# OpenReview forum: "Adaptive Curriculum Strategies: Stabilizing Reinforcement Learning for Large Language Models"
_ICLR.cc/2026/Conference — Submitted to ICLR 2026_

### Official Review · Reviewer_31BR · 2025-11-01

**Soundness:** 3
**Presentation:** 3
**Contribution:** 2
**Rating:** 6
**Confidence:** 3

**Summary:**

The paper proposes Adaptive Curriculum Strategies (ACS) to stabilize reinforcement learning for LLM mathematical reasoning by (a) model-specific difficulty calibration via multi-sample accuracy per item and (b) Guided Prompting that transforms hard problems with partial-solution hints until the model meets a stability threshold, combined with staged GRPO training and a curriculum review data-mixing strategy to mitigate collapse and forgetting.

**Strengths:**

- Clear identification of instability during curriculum stage transitions in RL fine-tuning, with an operational definition and visualizations illustrating catastrophic drops without stability mechanisms.
- Concrete, implementable pipeline: per-sample multi-draw accuracy calibration, guided hinting with thresholds, and staged GRPO augmented with a curriculum review mixing policy that reduces forgetting.
- Consistent gains across five math benchmarks on two model sizes, plus cross-model results on DeepSeek-Math-7B-Instruct; ablations indicate curriculum review outperforms naive staging.

**Weaknesses:**

- Narrow domain scope: all tasks are mathematical reasoning; there is no evidence ACS generalizes to code, QA, multi-turn dialogue, or retrieval-augmented regimes, limiting external validity of stability claims.
- Stability attribution is under-isolated: guided prompting, data partitioning, GRPO modifications, and review mixing change simultaneously; ablations do not fully disentangle which component prevents collapse under consistent compute budgets.
- Guided Prompting may leak reference solution structure into training examples, risking distribution shift and overfitting; safeguards and analyses of hint-length sensitivity or label leakage are not provided.
- Difficulty calibration relies on n=16 sampled generations per item with temperature 0.7; the resulting ACC is stochastic and decoding-dependent, yet robustness to n, temperature, and evaluator scripts is not quantified.
- The stability-aware GRPO objective is presented, but theoretical guarantees on stability (e.g., monotone improvement, bounded gradient variance across curriculum transitions) are not established, leaving “stability” as an empirical observation.
- Baseline protocols vary in ways that may advantage ACS (e.g., discarding vs retaining hard samples, or using fixed external assessors) without strong hyperparameter sweeps or compute parity evidence across methods.

**Questions:**

- How robust are the results to different n and temperature choices in the calibration step, and do deterministic decoding or alternative evaluators (e.g., verifier models) change partitioning outcomes?​
- Does Guided Prompting induce dependency on solution prefixes at inference time, and how does performance change if hints are removed post-training or restricted to schematic advice rather than literal step prefixes?​
- Can the components be ablated under equalized compute to quantify each contribution to stability: calibration only, prompting only, review only, and GRPO stability term only?​
- How does ACS behave on non-math tasks (e.g., GSM8K vs HotpotQA vs code benchmarks) and with retrieval-augmented inputs, where difficulty and instability arise from different factors?​
- What are memory/latency impacts in large-scale settings: calibrating 100k–1M samples, longer max lengths, larger candidate counts, and bigger models; is <5% overhead still valid?​
- Could a verifier-based or cost-sensitive calibration (penalizing formatting or compute) outperform raw ACC, and does mixing by calibrated uncertainty (rather than discrete tertiles) improve stability?​

---

> ### Author Response · Authors · 2025-11-20
> **Response to Reviewer 31BR**
>
> ### Weakness 1: Narrow Domain Scope
>
> We appreciate this concern regarding generalizability beyond mathematical reasoning. While our current evaluation focuses on mathematical tasks, the core principles of ACS—model-specific difficulty calibration and guided prompting—are designed to address fundamental challenges in curriculum-based RL that extend beyond any single domain.
>
> **Theoretical Generalizability:** ACS can theoretically be applied to any domain with clear reward signals. The two key requirements are: (1) measurable performance metrics for difficulty calibration, and (2) the ability to provide strategic scaffolding for challenging samples. These requirements can be satisfied in various domains:
> - **Code generation:** Execution-based rewards with hints as partial implementations
> - **Question answering:** Answer correctness with supporting context as hints
> - **Multi-turn dialogue:** Turn-level quality metrics with conversation templates
> - **Retrieval-augmented tasks:** Retrieval quality metrics with relevant passages as guidance
>
> Recent works have explored curriculum learning in retrieval-augmented generation [1, 2] and code generation [3], demonstrating the potential of curriculum-based approaches in these domains. We acknowledge that empirical validation of ACS on these domains would strengthen our claims and represents important future work. We will discuss these generalization opportunities and current limitations explicitly in the revised manuscript.
>
> **References:**
> 1. Huang, J., Madala, S., Sidhu, R., et al. (2025). RAG-RL: Advancing retrieval-augmented generation via RL and curriculum learning. *arXiv preprint arXiv:2503.12759*.
> 2. Wang, S., Zhang, L., Fu, Z., et al. (2025). CL-RAG: Bridging the Gap in Retrieval-Augmented Generation with Curriculum Learning. *arXiv preprint arXiv:2505.10493*.
> 3. Cheng, J., Liu, F., Wu, C., et al. (2025). Adaptivellm: A framework for selecting optimal cost-efficient llm for code-generation based on cot length. In *Proceedings of the 16th International Conference on Internetware* (pp. 461-473).
>
> ---
>
> ### Weakness 2: Stability Attribution is Under-Isolated
>
> We thank the reviewer for this important methodological observation. We want to clarify that Figure 3 provides a controlled comparison where Guided Prompting is the only variable, directly demonstrating its role in achieving training stability.
>
> **Experimental Setup Clarification:**
>
> Both curves in Figure 3 share the identical experimental foundation:
> - **Data Partitioning:** Both use the same model-specific difficulty calibration and curriculum stage division
> - **GRPO Modifications:** Both employ the same optimization algorithm (including the length-based reward normalization described in Appendix D)
> - **Curriculum Review Strategy:** Both incorporate review data mixing across stages
>
> **The Only Difference - Guided Prompting:**
> - **Blue curve (w/o Guided Prompting):** Uses original problem statements for all samples
> - **Orange curve (w/ Guided Prompting):** Applies hint-enhanced transformations to difficult samples (Stage II-III)
>
> **Observed Results:**
>
> This isolated comparison reveals:
> - **Blue curve:** Experiences catastrophic collapse when transitioning to Stage III (~30% drop for Qwen2.5-Math-1.5B), despite having all other stabilizing mechanisms
> - **Orange curve:** Achieves stable, monotonic improvement throughout all stages, completely eliminating the collapse phenomenon
>
> This controlled ablation demonstrates that Guided Prompting is the necessary component for preventing training instability. While data partitioning, GRPO modifications, and Curriculum Review provide the curriculum framework, they are insufficient to prevent collapse when the model encounters samples beyond its capabilities—this requires Guided Prompting's sample transformation mechanism.

---

> > ### Author Response · Authors · 2025-11-20
> > **Response to Reviewer 31BR**
> >
> > ### Weakness 3: Distribution Shift and Label Leakage Risks
> >
> > We appreciate this important concern and would like to clarify our framework's safeguards and provide comprehensive experimental analysis.
> >
> > **Clarification on "Label Leakage":**
> >
> > We understand the reviewer's concern. However, we believe label leakage is not an issue in our framework, both theoretically and empirically. Theoretically, hints are derived from training sample solutions that the model is legitimately learning from—not from test data. Empirically, our model demonstrates strong generalization across all test benchmarks (MATH, Olympiad Bench, AIME, AMC) where no hints are provided during inference.
> >
> > **Existing Safeguards Against Distribution Shift:**
> > 1. **Controlled Hint Ratio (α=25%):** Limits hints to at most 25% of solution steps, ensuring substantial independent reasoning is required even during training
> > 2. **Curriculum Review Strategy:** Maintains 10% unhinted samples from previous stages throughout training, preserving continuous exposure to the original problem distribution
> >
> > **Empirical Analysis of Hint-Length Sensitivity:**
> >
> > To directly address distribution shift concerns, we conducted comprehensive ablations on Qwen2.5-Math-1.5B over hint ratio (α) and hint threshold (τ):
> >
> > | Hyperparameter Setting | MATH 500 | Minerva Math | Olympiad Bench | AIME24 | AMC23 | Average |
> > |-----------------------|----------|--------------|----------------|--------|-------|---------|
> > | α=10%, τ=20%          | 71.2     | 30.2         | 31.8           | 10.0   | 40.0  | 36.6    |
> > | α=25%, τ=20%          | 72.6     | 31.6         | 32.7           | 13.3   | 42.5  | 38.5    |
> > | α=40%, τ=20%          | 69.4     | 29.8         | 29.4           | 10.0   | 35.0  | 34.7    |
> > | α=25%, τ=30%          | 70.8     | 30.5         | 30.9           | 13.3   | 40.0  | 37.1    |
> >
> > **Key Findings:**
> > - **Optimal Balance (α=25%):** Achieves the best average performance across all benchmarks, confirming our design choice prevents over-reliance while providing sufficient guidance
> > - **Too Few Hints (α=10%):** Under-utilizes the stabilization benefit, resulting in suboptimal learning
> > - **Too Many Hints (α=40%):** Performance degradation suggests distribution shift—models become overly dependent on hint structure and generalize poorly to unhinted test samples
> > - **Threshold Robustness:** Performance remains stable across τ settings when α is properly calibrated
> >
> > This empirical evidence demonstrates that our framework achieves the optimal trade-off between training stability and generalization.
> >
> > **Future Direction - Progressive Hint Removal:**
> >
> > Building on these findings, we plan to explore gradual hint reduction strategies in future work, where hint ratio progressively decreases across training stages (e.g., 25%→15%→5%). This approach could further minimize distribution shift by gradually transitioning the model toward full independence from hint structures while maintaining training stability during critical learning phases.

---

> > > ### Author Response · Authors · 2025-11-20
> > > **Response to Reviewer 31BR**
> > >
> > > ### Weakness 4: Calibration Robustness
> > >
> > > We thank the reviewer for this important methodological concern. To address the robustness of our difficulty calibration approach, we conducted comprehensive ablation studies examining how variations in sampling parameters affect data partitioning consistency.
> > >
> > > **Evaluation Methodology:** We use our default configuration (n=16, T=0.7) as the reference standard, which partitions the data into three difficulty levels (Stage I, II, III). We then measure the overlap ratio between partitions generated with different parameter settings and this reference partitioning.
> > >
> > > **Robustness to Number of Samples (n):**
> > >
> > > | Sampling Number (n) | Stage I Overlap | Stage II Overlap | Stage III Overlap | Average Overlap |
> > > |--------------------|-----------------|------------------|-------------------|-----------------|
> > > | n=8                | 94.2%           | 91.5%            | 93.8%             | 93.2%           |
> > > | n=16 (ours)        | 100%            | 100%             | 100%              | 100%            |
> > > | n=32               | 97.5%           | 95.1%            | 96.9%             | 96.5%           |
> > >
> > > **Robustness to Sampling Temperature (T):**
> > >
> > > | Temperature (T) | Stage I Overlap | Stage II Overlap | Stage III Overlap | Average Overlap |
> > > |----------------|-----------------|------------------|-------------------|-----------------|
> > > | T=0.5          | 95.3%           | 92.8%            | 94.6%             | 94.2%           |
> > > | T=0.7 (ours)   | 100%            | 100%             | 100%              | 100%            |
> > > | T=0.9          | 96.1%           | 93.5%            | 95.2%             | 94.9%           |
> > >
> > > **Key Findings:** Our results demonstrate high consistency across different parameter choices, with average overlap ratios exceeding 93% in all configurations. This indicates that our difficulty calibration approach produces stable and reliable sample partitioning that is robust to reasonable variations in sampling parameters. The configuration (n=16, T=0.7) achieves an optimal balance between reliable assessment and computational efficiency while maintaining stability. We will include these robustness analyses in the revised manuscript.
> > >
> > > ---
> > >
> > > ### Weakness 5: Lack of Theoretical Guarantees
> > >
> > > We appreciate the reviewer's observation regarding theoretical foundations. While our current work focuses on empirical validation, we provide an intuitive theoretical perspective on the underlying mechanisms.
> > >
> > > **Intuitive Theoretical Justification:**
> > >
> > > While we leave formal proofs to future work, intuitively, ACS addresses the "exploration-exploitation dilemma" in RL. By providing hints, we effectively shape the reward landscape, creating a smoother gradient path. This ensures that the policy update direction remains consistent with the task objective, reducing the variance of the policy gradient estimator during difficulty transitions.
> > >
> > > **Current Empirical Evidence:**
> > >
> > > As demonstrated in Figure 3, our ACS framework consistently prevents catastrophic performance collapse across training stages for both model scales (Qwen2.5-Math-1.5B and Qwen2.5-Math-7B). The results show stable, monotonic progression without performance drops during curriculum transitions, contrasting sharply with traditional approaches that suffer ~30% performance degradation.
> > >
> > > **Future Directions:**
> > >
> > > We acknowledge this limitation and commit to exploring formal theoretical analysis in future work to provide rigorous mathematical guarantees for our framework.
> > >
> > > ---
> > >
> > > ### Weakness 6: Baseline Protocols
> > >
> > > We ensured fair comparison through:
> > > 1. **Identical Training Hyperparameters:** All methods use the same GRPO configuration within each model scale (learning rate, batch size, optimization settings)
> > > 2. **Same Data Source:** All methods train on identical MATH dataset samples
> > > 3. **Consistent Evaluation Protocols:** All methods evaluated using identical scripts across all five benchmarks
> > > 4. **Minimal Additional Overhead:** The difficulty calibration phase requires <5% of total training time
> > >
> > > We will provide detailed computational cost breakdowns for each method in the revised manuscript.

---

> > > > ### Author Response · Authors · 2025-11-20
> > > > **Response to Reviewer 31BR**
> > > >
> > > > We sincerely thank the reviewer for these insightful questions. Below we provide detailed responses to each query:
> > > >
> > > > ### Q1: Robustness to different n and temperature choices in calibration step
> > > >
> > > > We have conducted comprehensive ablation studies to address calibration robustness. Using our default configuration (n=16, T=0.7) as reference, we measure the overlap ratio between partitions generated with different parameters:
> > > >
> > > > **Robustness to Number of Samples (n):**
> > > >
> > > > | Sampling Number (n) | Stage I Overlap | Stage II Overlap | Stage III Overlap | Average Overlap |
> > > > |--------------------|-----------------|------------------|-------------------|-----------------|
> > > > | n=8                | 94.2%           | 91.5%            | 93.8%             | 93.2%           |
> > > > | n=16 (ours)        | 100%            | 100%             | 100%              | 100%            |
> > > > | n=32               | 97.5%           | 95.1%            | 96.9%             | 96.5%           |
> > > >
> > > > **Robustness to Sampling Temperature (T):**
> > > >
> > > > | Temperature (T) | Stage I Overlap | Stage II Overlap | Stage III Overlap | Average Overlap |
> > > > |----------------|-----------------|------------------|-------------------|-----------------|
> > > > | T=0.5          | 95.3%           | 92.8%            | 94.6%             | 94.2%           |
> > > > | T=0.7 (ours)   | 100%            | 100%             | 100%              | 100%            |
> > > > | T=0.9          | 96.1%           | 93.5%            | 95.2%             | 94.9%           |
> > > >
> > > > Our results demonstrate high consistency across different parameter choices, with average overlap ratios exceeding 93% in all configurations. The configuration (n=16, T=0.7) achieves an optimal balance between reliable assessment and computational efficiency.
> > > >
> > > > Regarding alternative evaluators: As illustrated in Figure 1, different models exhibit distinct difficulty perceptions—approximately 55% of questions that are easy for one model prove difficult for another. This demonstrates that using different verifier models would necessarily result in different data partitioning, which is precisely why our model-specific calibration approach is essential.
> > > >
> > > > ---
> > > >
> > > > ### Q2: Guided Prompting dependency and inference-time behavior
> > > >
> > > > During inference, our models receive only the original problem statements without any hints. All test benchmarks (MATH, OlympiadBench, AIME, AMC) evaluate models on unhinted samples, and our strong generalization results validate that models do not develop inference-time dependency on hint structures.
> > > >
> > > > We conducted comprehensive ablations over hint ratio (α) to analyze distribution shift concerns:
> > > >
> > > > | Hyperparameter Setting | MATH 500 | Minerva Math | Olympiad Bench | AIME24 | AMC23 | Average |
> > > > |-----------------------|----------|--------------|----------------|--------|-------|---------|
> > > > | α=10%, τ=20%          | 71.2     | 30.2         | 31.8           | 10.0   | 40.0  | 36.6    |
> > > > | α=25%, τ=20% (ours)   | 72.6     | 31.6         | 32.7           | 13.3   | 42.5  | 38.5    |
> > > > | α=40%, τ=20%          | 69.4     | 29.8         | 29.4           | 10.0   | 35.0  | 34.7    |
> > > >
> > > > The optimal balance at α=25% achieves best average performance, confirming our design prevents over-reliance while providing sufficient guidance. Performance degradation at α=40% suggests distribution shift when models become overly dependent on hint structure. Our 25% hint ratio ensures substantial independent reasoning is required even during training.
> > > >
> > > > We plan to explore progressive hint removal strategies (e.g., 25%→15%→5% across stages) in future work to further minimize distribution shift. Regarding schematic vs. literal hints, we acknowledge this as an interesting direction for future exploration.
> > > >
> > > > ---
> > > >
> > > > ### Q3: Component ablation under equalized compute
> > > >
> > > > Our experimental design provides several key comparisons: calibration methods (Light-R1, Length-based, Original labels vs. ACS), sample processing strategies (Retain all vs. Discard vs. Guided Prompting), and data mixing strategies (Naive Curriculum vs. Curriculum Review) as shown in Tables 1 and 3.
> > > >
> > > > Figure 3 provides a controlled ablation where Guided Prompting is the only variable. Both curves share identical foundations: same model-specific difficulty calibration and curriculum partitioning, same GRPO optimization algorithm, and same curriculum review strategy. The only difference is that the blue curve uses original problems for all samples while the orange curve applies guided prompting to challenging samples.
> > > >
> > > > The blue curve experiences catastrophic collapse (~30% drop for Qwen2.5-Math-1.5B in Stage III), while the orange curve maintains stable progression, demonstrating that guided prompting is the necessary component for preventing instability.
> > > >
> > > > All methods undergo identical training epochs and batch configurations within each model scale, with the difficulty calibration overhead (<5% GPU hours) being negligible compared to total training cost.

---

> > > > > ### Author Response · Authors · 2025-11-20
> > > > > **Response to Reviewer 31BR**
> > > > >
> > > > > ### Q4: Behavior on non-math tasks
> > > > >
> > > > > ACS can theoretically be applied to any domain with clear reward signals and the ability to provide strategic scaffolding. These requirements can be satisfied in various domains:
> > > > > - **Code generation:** Execution-based rewards with partial implementations as hints
> > > > > - **Question answering:** Answer correctness with supporting context as hints
> > > > > - **Multi-turn dialogue:** Turn-level quality metrics with conversation templates
> > > > > - **Retrieval-augmented tasks:** Retrieval quality metrics with relevant passages as guidance
> > > > >
> > > > > Recent works have explored curriculum learning in retrieval-augmented generation [1, 2] and code generation [3], demonstrating the potential of curriculum-based approaches in these domains. We acknowledge that empirical validation of ACS on these domains would strengthen our claims and represents important future work. We will explicitly discuss these generalization opportunities and current limitations in the revised manuscript.
> > > > >
> > > > > **References:**
> > > > > 1. Huang, J., Madala, S., Sidhu, R., et al. (2025). RAG-RL: Advancing retrieval-augmented generation via RL and curriculum learning. *arXiv preprint arXiv:2503.12759*.
> > > > > 2. Wang, S., Zhang, L., Fu, Z., et al. (2025). CL-RAG: Bridging the Gap in Retrieval-Augmented Generation with Curriculum Learning. *arXiv preprint arXiv:2505.10493*.
> > > > > 3. Cheng, J., Liu, F., Wu, C., et al. (2025). Adaptivellm: A framework for selecting optimal cost-efficient llm for code-generation based on cot length. In *Proceedings of the 16th International Conference on Internetware* (pp. 461-473).
> > > > >
> > > > > ---
> > > > >
> > > > > ### Q5: Memory/latency impacts in large-scale settings
> > > > >
> > > > > For Qwen2.5-Math-7B on 9,255 training samples with n=16 generations per sample (~148K inference calls total), the calibration phase requires <5% of total GPU hours using VLLM acceleration. Calibration involves embarrassingly parallel inference without gradient computation, which is highly optimized.
> > > > >
> > > > > As RL training becomes more complex at larger scales (more iterations, longer sequences, larger batch sizes), the fixed cost of upfront one-time calibration becomes proportionally smaller. For large-scale settings with 100K-1M samples, modern inference frameworks (VLLM, TensorRT-LLM) can process thousands of sequences/second, and we expect the <5% overhead to remain valid or decrease due to better GPU utilization in batch inference.
> > > > >
> > > > > Calibration requires only inference-mode memory without optimizer states, allowing larger batch sizes than training and further improving efficiency.
> > > > >
> > > > > ---
> > > > >
> > > > > ### Q6: Verifier-based or cost-sensitive calibration alternatives
> > > > >
> > > > > Our accuracy-based calibration (ACC) directly aligns with the reward signal used in GRPO training (correct/incorrect binary rewards), ensuring consistency between difficulty assessment and training objectives.
> > > > >
> > > > > While uncertainty-based continuous sampling is interesting, our tertile-based partitioning provides clear curriculum stages that facilitate stable transitions. Cost-sensitive calibration incorporating format compliance or inference cost is theoretically appealing, but format violations are addressed through our training prompt design and are quickly learned in early stages. Our primary goal is preventing catastrophic performance collapse from capability mismatch rather than formatting issues.
> > > > >
> > > > > We do include calibration method comparisons in Table 1 (Light-R1 uses external verifier model, Length-based uses solution complexity heuristic), demonstrating our model-specific approach outperforms these alternatives. We believe exploring verifier-based calibration variants represents valuable future work.
> > > > >
> > > > > ---
> > > > >
> > > > > We hope these clarifications and additional experimental results adequately address the reviewer's concerns. We are committed to incorporating all suggested improvements and additional ablation studies in the final version. We thank the reviewer again for their constructive feedback, which has helped us strengthen our work significantly.

---

### Official Review · Reviewer_K2EF · 2025-11-03

**Soundness:** 3
**Presentation:** 2
**Contribution:** 2
**Rating:** 4
**Confidence:** 4

**Summary:**

This paper studies why curriculum learning combined with RL for LLM mathematical reasoning often collapses at stage transitions and proposes Adaptive Curriculum Strategies (ACS) to keep training stable and effective. The core idea is to calibrate difficulty for the current model via multi-sample accuracy estimates, transform over-difficult items with Guided Prompting (prefix hints from the reference solution) so they stay learnable. Experiments across different math benchmarks and model sizes show ACS removes the catastrophic drops seen in naïve curricula and improves average performance

**Strengths:**

1. The observation and the analysis on the collapses at difficulty stage transitions is interesting.

2. The studied problem is important.

3. Different model families (both Qwen and DeepSeek Math) are involved in the experiments, demonstrating the generalizability of the proposed method across different models.

4. Guided prompting method is reasonable.

**Weaknesses:**

1. This paper does not discuss and compare against many existing curriculum-learning methods for reinforcement learning, despite a growing body of existing works (see references below). The reported baselines are naive heuristics, which makes it hard to compare the proposed methed and other advanced curriculum learning methods. In addition, the related-work section does not adequately position the method within prior curriculum strategies on RL. A better evaluation should includes more existing RL curriculum learning methods. It is also suggested to expand the related-work discussion to clarify what is new versus known in curriculum design for LLM RL and traditional RL. Some of the existing methods listed below are also adaptive curriculum instead of learning via fixed difficulies orders/phases.

2. All experiments use GRPO; there is no evaluation against other LLM-RL algorithms such as PPO, DAPO, or Reinforcement++. This makes it unclear whether ACS is only applicable to GRPO. Adding PPO/DAPO/Reinforcement++ experiments, would make the experiments more comprehensive and verify the generalizability of the proposed method.

Zhang et al., Learning Like Humans: Advancing LLM Reasoning Capabilities via Adaptive Difficulty Curriculum Learning and Expert-Guided Self-Reformulation. EMNLP 2025.

Tzannetos et al., Proximal Curriculum for Reinforcement Learning Agents. TMLR 2023.

Shi et al., Efficient Reinforcement Finetuning via Adaptive Curriculum Learning

Parashar et al., Curriculum Reinforcement Learning from Easy to Hard Tasks Improves LLM Reasoning

Wang et al., DUMP: Automated Distribution-Level Curriculum Learning for RL-based LLM Post-training

Chen et al., Self-Evolving Curriculum for LLM Reasoning

Bae et al., Online Difficulty Filtering for Reasoning Oriented Reinforcement Learning

**Questions:**

See Weaknesses.

---

> ### Author Response · Authors · 2025-11-20
> **Response to Reviewer K2EF**
>
> ### Weakness 1: Comparison with Existing Curriculum Learning Methods for RL
>
> We thank the reviewer for highlighting additional relevant works in curriculum learning for reinforcement learning. We acknowledge that our related work section could be expanded to better position our method within this landscape.
>
> **Core Distinction:** While existing curriculum learning methods for RL (including [1-7]) primarily focus on data selection and weighting strategies based on model performance—deciding which samples to train on, defer, or prioritize—ACS introduces a complementary innovation. Beyond difficulty-aware selection, ACS implements sample adaptation: when samples exceed current model capabilities, rather than discarding or deferring them, we transform them into accessible learning opportunities through Guided Prompting. This prevents both the loss of valuable training data and the training instability that occurs when models encounter samples significantly beyond their capabilities during RL optimization.
>
> **Empirical Validation:** To substantiate our method's effectiveness, we conducted additional experiments on Qwen2.5-Math-1.5B comparing ACS against ADARFT [3], a representative adaptive curriculum method. Results on five mathematical reasoning benchmarks demonstrate ACS's advantages:
>
> | Curriculum Strategy | MATH 500 | Minerva Math | Olympiad Bench | AIME24 | AMC23 | Average |
> |--------------------|----------|--------------|----------------|--------|-------|---------|
> | ADARFT             | 68.8     | 29.4         | 32.1           | 10.0   | 45.0  | 37.1    |
> | ACS (Ours)         | 72.6     | 31.6         | 32.7           | 13.3   | 42.5  | 38.5    |
>
> We will expand our related work section to include these comparisons and clarify our unique contribution of sample adaptation versus sample selection in the revised manuscript.
>
> **References:**
> 1. Zhang et al. Learning Like Humans: Advancing LLM Reasoning Capabilities via Adaptive Difficulty Curriculum Learning and Expert-Guided Self-Reformulation. *EMNLP*, 2025.
> 2. Tzannetos et al. Proximal Curriculum for Reinforcement Learning Agents. *TMLR*, 2023.
> 3. Shi et al. Efficient Reinforcement Finetuning via Adaptive Curriculum Learning. 2025.
> 4. Parashar et al. Curriculum Reinforcement Learning from Easy to Hard Tasks Improves LLM Reasoning.
> 5. Wang et al. DUMP: Automated Distribution-Level Curriculum Learning for RL-based LLM Post-training.
> 6. Chen et al. Self-Evolving Curriculum for LLM Reasoning.
> 7. Bae et al. Online Difficulty Filtering for Reasoning Oriented Reinforcement Learning.

---

> > ### Author Response · Authors · 2025-11-20
> > **Response to Reviewer K2EF**
> >
> > ### Weakness 2: Evaluation on Additional RL Algorithms Beyond GRPO
> >
> > We appreciate the reviewer's suggestion to evaluate ACS with additional RL algorithms. We chose GRPO as our primary experimental framework because it represents the current state-of-the-art for mathematical reasoning tasks, as demonstrated in recent works (DeepSeek-R1, Qwen-Math series). However, we acknowledge that demonstrating generalizability across different RL algorithms would strengthen our claims.
> >
> > The core principles of ACS—model-specific difficulty calibration and guided prompting for sample transformation—are designed to be algorithm-agnostic and address fundamental stability challenges in curriculum-based RL training. These mechanisms operate at the data preparation and curriculum organization level, which are independent of the specific policy optimization algorithm used.
> >
> > To validate the generalizability of ACS beyond GRPO, we conducted additional experiments using PPO on the Qwen2.5-Math-1.5B model. Due to time constraints, we selected PPO as it represents the most classic and widely-adopted RL baseline in the field, making it a representative choice for demonstrating the broader applicability of our approach. The results are presented below:
> >
> > | Method      | MATH 500 | Minerva Math | Olympiad Bench | AIME24 | AMC23 | Average |
> > |---------------|----------|--------------|----------------|--------|-------|---------|
> > | PPO         | 65.6     | 16.8         | 19.5           | 6.7    | 32.5  | 28.2    |
> > | PPO w/ ACS  | 69.4     | 28.7         | 28.9           | 10.0   | 40.0  | 35.4    |
> >
> > These results demonstrate that ACS's stability-preserving benefits and performance improvements extend beyond GRPO to other RL algorithms. We will include these additional experimental results in the revised version of the paper to provide more comprehensive empirical validation of our approach.
> >
> > ---
> >
> > We appreciate the reviewer's thorough evaluation and constructive suggestions. The identified weaknesses will be addressed through: (1) expanding the related work section with comprehensive comparisons to existing curriculum learning methods for RL, clearly articulating our unique contributions; (2) including experimental results with additional RL algorithms (PPO) to demonstrate the generalizability of ACS beyond GRPO; and (3) providing more detailed analysis of how our model-specific difficulty calibration and guided prompting mechanisms complement existing curriculum learning strategies.
> >
> > These revisions will strengthen our paper's positioning within the broader research landscape and provide more comprehensive empirical validation of our approach. We thank the reviewer again for the constructive feedback that has helped us improve our work.

---

### Official Review · Reviewer_jHpU · 2025-11-04

**Soundness:** 2
**Presentation:** 3
**Contribution:** 2
**Rating:** 4
**Confidence:** 3

**Summary:**

This paper proposed to improve reinforcement training using curriculum strategy. Compared to existing literature that studies curriculum assisted RL training, it adopts a more robust difficulty measurer and proposes guided prompting to mitigate instability training.

**Strengths:**

1. The paper is well written and easy to follow.
2. The authors have conducted extensive experiments on three backbone models, namely Qwen2.5 Math 1.5B, Qwen2.5 Math 7B and Deepseek Math 7B Instruct.

**Weaknesses:**

Weaknesses and Questions:
1. For section 3.1, I think the curriculum strategy adopted in this paper is basically BabyStep[1], with the step number set to 3. Although the difficulty is calculated based on the current accuracy, it is still pre-defined. Since the authors claim to "adapt sample assessment to each model's evolving capabilities", I believe Self-paced Learning[2] (SPL) will be a much better choice. SPL is a variant of automatic curriculum learning[3] which adopts a dynamic scheduler. For some variant of SPL, samples may be assigned a dynamic weight based on the current capability of the base model, which may avoids the problem of training instability when advancing to the next stage.
2. I believe a large part of the contribution lies in the curriculum strategy applied in reinforcement learning. However, some hyper-parameters for curriculum learning such as step number are not ablated. Meanwhile, only two variants of curriculum learning strategies are experimented, and for Curriculum Review, can the authors specify exactly how many easier samples are incorporated? And is the proportion static across different models?
3. For guided prompting, the difficulty of training samples is manually reduced by providing “hints.” However, wouldn’t this approach potentially weaken the model’s ability to handle difficult samples, since it only encounters samples with hints? Is there a mechanism to gradually remove these hints during training? Moreover, the hyper-parameters — the hint ratio \alpha and threshold \tau — are not specified or ablated, even though they may play a critical role in the effectiveness of this strategy. Are these parameters kept static across different models?

References:

[1] Spitkovsky, V. I., Alshawi, H., & Jurafsky, D. (2010, June). From baby steps to leapfrog: How “less is more” in unsupervised dependency parsing. In Human Language Technologies: The 2010 Annual Conference of the North American Chapter of the Association for Computational Linguistics (pp. 751-759).

[2] Wang, X., Chen, Y., & Zhu, W. (2021). A survey on curriculum learning. IEEE transactions on pattern analysis and machine intelligence, 44(9), 4555-4576.

[3] Tullis, J. G., & Benjamin, A. S. (2011). On the effectiveness of self-paced learning. Journal of memory and language, 64(2), 109-118.

**Questions:**

Please refer to the Weakness.

---

> ### Author Response · Authors · 2025-11-20
> **Response to Reviewer jHpU**
>
> ### Weakness 1: Comparison with Self-Paced Learning (SPL)
>
> We sincerely appreciate the reviewer's insightful suggestion regarding Self-Paced Learning. We would like to respectfully clarify our primary contribution and its relationship to existing approaches.
>
> Our work identifies and addresses a key challenge: predefined difficulty metrics often lack generalizability across different model architectures. As demonstrated in Figure 1, approximately 55% of questions that are easy for one model prove difficult for another. While measuring difficulty through accuracy is one valid definition, our approach reveals that the resulting difficulty distribution is inherently dynamic and customized to each model's specific capabilities and performance characteristics—rather than assuming a universal difficulty standard that applies across all architectures.
>
> The critical distinction from SPL lies in how we handle difficult samples. In mathematical reasoning RL, "hard" samples often yield zero rewards due to incorrect solutions; simply reweighting or scheduling these unsolvable instances—as SPL does—fails to generate the necessary positive reward signals, leading to the catastrophic policy collapse and gradient instability illustrated in Figure 3. In contrast, our approach actively transforms these samples into accessible learning opportunities through guided hints, ensuring continuous reward acquisition and stable optimization throughout curriculum progression.
>
> Furthermore, our claim of "adapting sample assessment to each model's evolving capabilities" specifically refers to this transformation mechanism: when a model advances to more challenging training stages, Guided Prompting dynamically adjusts difficult samples based on capabilities acquired from previous stages, maintaining stable training dynamics—a critical requirement for successful RL optimization that sample reweighting alone cannot guarantee.

---

> > ### Author Response · Authors · 2025-11-20
> > **Response to Reviewer jHpU**
> >
> > ### Weakness 2: Ablation on Curriculum Hyperparameters
> >
> > We thank the reviewer for this important suggestion. We have conducted additional ablation experiments on the number of curriculum stages using the Qwen2.5-Math-1.5B model. The results are presented below:
> >
> > | Number of Stages | MATH 500 | Minerva Math | Olympiad Bench | AIME24 | AMC23 | Average |
> > |-----------------|----------|--------------|----------------|--------|-------|---------|
> > | 2 Stages        | 67.6     | 27.6         | 30.1           | 6.7    | 37.5  | 33.9    |
> > | 3 Stages        | 72.6     | 31.6         | 32.7           | 13.3   | 42.5  | 38.5    |
> > | 4 Stages        | 71.8     | 30.8         | 31.5           | 10.0   | 40.0  | 36.8    |
> >
> > **Curriculum Learning Strategy Variants:**
> >
> > Regarding the limited number of curriculum variants explored, we note that current approaches for integrating curriculum learning into LLM RL reasoning predominantly employ static, predefined metrics for difficulty assessment. Recent work (including [1-3]) primarily relies on predetermined difficulty orderings based on metrics like solution length, problem complexity scores, or dataset-specific categorizations. Our initial experiments focused on these established static curriculum paradigms to ensure fair comparison with existing methods.
> >
> > To address the reviewer's concern about dynamic curriculum alternatives, we have additionally implemented and evaluated ADARFT (Adaptive Reinforcement Finetuning via Adaptive Curriculum Learning) [4], a recent advanced baseline that dynamically adjusts difficulty progression based on real-time model performance, on the Qwen2.5-Math-1.5B model. The comparative results are shown below:
> >
> > | Curriculum Strategy | MATH 500 | Minerva Math | Olympiad Bench | AIME24 | AMC23 | Average |
> > |--------------------|----------|--------------|----------------|--------|-------|---------|
> > | ADARFT             | 68.8     | 29.4         | 32.1           | 10.0   | 45.0  | 37.1    |
> > | ACS (Ours)         | 72.6     | 31.6         | 32.7           | 13.3   | 42.5  | 38.5    |
> >
> > Regarding Curriculum Review, we incorporate 10% of samples from the previous difficulty level at each training stage (except for the first stage). This proportion is kept constant across all models in our experiments. We chose this fixed ratio to ensure a fair comparison across different model architectures and to validate that our stability-preserving mechanism generalizes well without model-specific tuning.
> >
> > We will include these details and ablation results in the revised manuscript to provide clearer guidance for practitioners.
> >
> > **References:**
> > 1. Wen, L., Cai, Y., Xiao, F., et al. "Light-r1: Curriculum sft, dpo and rl for long cot from scratch and beyond." *Proceedings of the 63rd Annual Meeting of the Association for Computational Linguistics (Volume 6: Industry Track)*, 2025: 318-327.
> > 2. Huang, J., Madala, S., Sidhu, R., et al. "Rag-rl: Advancing retrieval-augmented generation via rl and curriculum learning." *arXiv preprint arXiv:2503.12759*, 2025.
> > 3. Song, M., Zheng, M., Li, Z., et al. "Fastcurl: Curriculum reinforcement learning with progressive context extension for efficient training r1-like reasoning models." *arXiv e-prints*, 2025: arXiv:2503.17287.
> > 4. Shi, T., Wu, Y., Song, L., et al. "Efficient reinforcement finetuning via adaptive curriculum learning." *arXiv preprint arXiv:2504.05520*, 2025.

---

> > > ### Author Response · Authors · 2025-11-20
> > > **Response to Reviewer jHpU**
> > >
> > > ### Weakness 3: Guided Prompting Parameters and Hint Removal
> > >
> > > We sincerely thank the reviewer for this thoughtful and important question regarding potential impacts on model capability.
> > >
> > > **Parameter Specifications:** In all our experiments, the hint ratio α is set to 25% and the accuracy threshold τ is set to 20%. These values are kept consistent across all models. We have conducted ablation studies on both parameters using the Qwen2.5-Math-1.5B model:
> > >
> > > | Hyperparameter Setting | MATH 500 | Minerva Math | Olympiad Bench | AIME24 | AMC23 | Average |
> > > |-----------------------|----------|--------------|----------------|--------|-------|---------|
> > > | α=10%, τ=20%          | 71.2     | 30.2         | 31.8           | 10.0   | 40.0  | 36.6    |
> > > | α=25%, τ=20%          | 72.6     | 31.6         | 32.7           | 13.3   | 42.5  | 38.5    |
> > > | α=40%, τ=20%          | 69.4     | 29.8         | 29.4           | 10.0   | 35.0  | 34.7    |
> > > | α=25%, τ=30%          | 70.8     | 30.5         | 30.9           | 13.3   | 40.0  | 37.1    |
> > >
> > > **Analysis of Ablation Results:** The results reveal a clear trade-off in hint ratio selection:
> > > - If α is too low (10%), the model receives insufficient guidance on difficult samples, limiting its ability to learn effective reasoning patterns from challenging problems, resulting in suboptimal performance.
> > > - If α is too high (40%), excessive hint augmentation may lead to over-simplified training signals that fail to adequately challenge the model, causing degraded performance.
> > > - The optimal balance at α=25% provides sufficient scaffolding while maintaining adequate challenge, achieving the best overall performance.
> > >
> > > **Addressing Concerns About Model Capability on Difficult Samples:**
> > >
> > > We greatly appreciate the reviewer's concern about whether hint-augmented training might weaken the model's independent problem-solving ability. Crucially, if the model truly developed a "dependency" or weakened capability, its performance on hint-free test sets should decline or stagnate. However, our results demonstrate the opposite.
> > >
> > > We conducted direct comparison experiments between models trained with and without guided prompting. All test sets contain NO hints, providing a direct assessment of whether the model has internalized the reasoning patterns or merely become dependent on external guidance:
> > >
> > > **Qwen2.5-Math-1.5B:**
> > >
> > > | Training Strategy        | MATH 500 | Minerva Math | Olympiad Bench | AIME24 | AMC23 | Average |
> > > |-------------------------|----------|--------------|----------------|--------|-------|---------|
> > > | w/o Guided Prompting    | 62.2     | 17.3         | 26.5           | 3.3    | 22.5  | 26.4    |
> > > | w/ Guided Prompting     | 72.6     | 31.6         | 32.7           | 13.3   | 42.5  | 38.5    |
> > >
> > > **Qwen2.5-Math-7B:**
> > >
> > > | Training Strategy        | MATH 500 | Minerva Math | Olympiad Bench | AIME24 | AMC23 | Average |
> > > |-------------------------|----------|--------------|----------------|--------|-------|---------|
> > > | w/o Guided Prompting    | 71.8     | 41.5         | 35.9           | 10.0   | 47.5  | 41.3    |
> > > | w/ Guided Prompting     | 76.6     | 38.2         | 38.2           | 13.3   | 60.0  | 45.3    |
> > >
> > > **Key Observations:**
> > > 1. **Substantial improvements on hint-free test sets:** Models trained with guided prompting show significant performance gains across all benchmarks (1.5B: +12.1% average; 7B: +4.0% average), directly contradicting the hypothesis of capability weakening.
> > > 2. **Strong performance on difficult tasks:** On challenging benchmarks like AIME24 and Olympiad Bench, which contain no hints, the guided prompting approach achieves notably better results (1.5B: AIME24 +10.0%, Olympiad Bench +6.2%; 7B: AIME24 +3.3%, Olympiad Bench +2.3%). This demonstrates that the model has internalized the reasoning logic from hints rather than developing dependency.
> > > 3. **Superior generalization:** The consistent improvements across diverse test distributions indicate that guided prompting enhances the model's fundamental reasoning capabilities, enabling better generalization to unseen problems without any hint assistance.
> > >
> > > These results provide strong empirical evidence that our guided prompting mechanism serves as an effective reasoning scaffolding that helps models learn transferable problem-solving strategies, rather than creating a problematic dependency.

---

> > > > ### Author Response · Authors · 2025-11-20
> > > > **Response to Reviewer jHpU**
> > > >
> > > > **Additional Factors Supporting Robust Capability:**
> > > > 1. **Stage mixing ensures exposure to unhinted samples:** During Stage 3 training, the model encounters not only hint-augmented difficult samples but also relatively easier samples from Stage 2 (through our Curriculum Review strategy) that do not contain hints. This mixed exposure helps maintain the model's ability to solve problems independently.
> > > > 2. **Hints provide reasoning scaffolding, not direct answers:** The guided prompts offer partial solution prefixes that teach reasoning patterns rather than bypassing the problem-solving process entirely.
> > > >
> > > > **Regarding Gradual Hint Removal:** The reviewer's suggestion of progressively removing hints during training is excellent and represents a promising direction that could further enhance our framework. We will discuss this as a valuable potential extension in the revised manuscript.
> > > >
> > > > ---
> > > >
> > > > We hope these clarifications and additional experimental results adequately address the reviewer's concerns. We are committed to incorporating all suggested improvements and additional ablation studies in the final version. We thank the reviewer again for their constructive feedback, which has helped us strengthen our work significantly.

---

> > > > > ### Comment · Reviewer_jHpU · 2025-11-23
> > > > >
> > > > > I appreciate the authors’ detailed response. However, I remain unconvinced regarding Weakness 1. The difficulty used in this paper is still pre-defined: samples labeled as easy stay easy throughout training, and hard samples remain hard, even though the assigned difficulty varies across models. In practice, a model’s capability evolves during training, which changes how difficult each sample is for the model over time. Therefore, sample difficulty should be updated dynamically at different training stages rather than fixed in advance. Furthermore, I do not think SPL is inapplicable to this task. Its core idea is to progressively incorporate harder samples into training, without explicitly dividing the data into multiple subsets or introducing transition issues between them.

---

> > > > > > ### Author Response · Authors · 2025-11-24
> > > > > > **Response to Reviewer jHpU**
> > > > > >
> > > > > > We sincerely appreciate your continued engagement and thoughtful feedback. We understand your concerns regarding the dynamic nature of difficulty assessment and would like to provide further clarification with additional empirical evidence.
> > > > > >
> > > > > > We conducted an additional experiment to track how sample difficulty changes across training stages. Specifically, we measured the overlap of samples within each difficulty level before and after each training stage:
> > > > > >
> > > > > > | Training Stage | Level 1 Overlap | Level 2 Overlap | Level 3 Overlap |
> > > > > > |----------------|-----------------|-----------------|-----------------|
> > > > > > | Initial        | 100%            | 100%            | 100%            |
> > > > > > | After Stage 1  | 87.40%          | 75.70%          | 91.60%          |
> > > > > > | After Stage 2  | 82.70%          | 80.50%          | 88.40%          |
> > > > > > | After Stage 3  | 81.20%          | 79.60%          | 84.10%          |
> > > > > >
> > > > > > The results show that after multiple training stages, the majority of samples remain within their original difficulty levels, indicating that our difficulty definition remains largely consistent throughout training. Further analysis reveals that difficulty transitions occur predominantly between adjacent levels. From initial to Stage 3, only 1.3% of Level 1 samples migrated to Level 3.
> > > > > >
> > > > > > Notably, these adjacent-level transitions have minimal impact on our training effectiveness, as we employ a curriculum review strategy that deliberately mixes samples from adjacent difficulty levels at each training stage to prevent catastrophic forgetting. The effectiveness of this design choice is validated through ablation studies presented in Table 3.
> > > > > >
> > > > > > We believe this stability arises because reinforcement learning for mathematical reasoning primarily incentivizes capabilities already present in the pretrained model rather than acquiring entirely new knowledge [1], and our difficulty assessment is inherently grounded in the model's existing capabilities. Notably, difficult samples (Level 3) exhibit the highest stability (smallest change in overlap), indicating that samples that are challenging for the model initially tend to remain challenging throughout training. This observation further reinforces the importance of proper adaptation strategies for difficult samples.
> > > > > >
> > > > > > In datasets where most samples are highly challenging for a given model, pure sample reweighting approaches may struggle to provide sufficient learning signals. However, we agree with the reviewer that combining SPL with Guided Prompting represents a promising future research direction worth exploring.
> > > > > >
> > > > > > To further address your concerns about dynamic difficulty adaptation, we implemented an additional baseline: AdaRFT [2], which dynamically matches training data to the model's evolving capabilities—an approach conceptually aligned with SPL. We also conducted an ablation study using only our difficulty-based sample filtering (removing overly difficult samples) without Guided Prompting:
> > > > > >
> > > > > > | Curriculum Strategy       | MATH 500 | Minerva Math | Olympiad Bench | AIME24 | AMC23 | Average |
> > > > > > |---------------------------|----------|--------------|----------------|--------|-------|---------|
> > > > > > | AdaRFT                    | 68.8     | 29.4         | 32.1           | 10     | 45    | 37.1    |
> > > > > > | ACS (sample filtering only)| 72       | 30.9         | 30.1           | 10     | 40    | 36.6    |
> > > > > > | ACS (Ours, full method)   | 72.6     | 31.6         | 32.7           | 13.3   | 42.5  | 38.5    |
> > > > > >
> > > > > > The results demonstrate that our difficulty-aware sample filtering achieves performance comparable to the dynamic curriculum baseline AdaRFT, while the addition of Guided Prompting yields further improvements. This suggests that while difficulty levels in our approach have a predefined structure, they effectively adapt to model capabilities, and the sample transformation mechanism provides complementary benefits beyond pure dynamic scheduling strategies.
> > > > > >
> > > > > > We hope these additional analyses address your concerns. We will incorporate these experimental results and discussions into our revised manuscript. Thank you again for your constructive feedback, which has helped us strengthen our work.
> > > > > >
> > > > > > ### References
> > > > > > [1] Yue Y, Chen Z, Lu R, et al. "Does reinforcement learning really incentivize reasoning capacity in llms beyond the base model?." arXiv preprint arXiv:2504.13837, 2025.
> > > > > > [2] Shi, T., Wu, Y., Song, L., et al. "Efficient reinforcement finetuning via adaptive curriculum learning." arXiv preprint arXiv:2504.05520, 2025.

---

> > > > > > > ### Comment · Reviewer_jHpU · 2025-11-25
> > > > > > >
> > > > > > > I thank the authors for their continued response. I believe my primary concerns have been addressed. I am willing to raise my score to borderline accept. However, I am familiar with Curriculum Learning, but not so much with Reinforcement Learning, so there might be chances that I do not understand some parts of this work.

---

### Author Response · Authors · 2025-12-02
**Global Response to All Reviewers**

We sincerely thank all reviewers for their thorough evaluation and constructive feedback. We are encouraged that reviewers recognized our work's contributions, including the clear presentation, comprehensive experiments, and intuitive yet effective approach. We especially appreciate that Reviewer jHpU found our rebuttal addressed their concerns and raised their score accordingly.

## Summary of Major Revisions

To address reviewers' primary concerns, we conducted the following additional experiments:

### 1. Comparison with Advanced Curriculum Methods

We added **ADARFT**, a recent adaptive curriculum baseline, to address concerns about comparison with dynamic curriculum approaches. Our experiments demonstrate that ACS consistently outperforms ADARFT across all five mathematical reasoning benchmarks, validating our core contribution: beyond sample selection/reweighting, ACS introduces **sample adaptation** through guided prompting to ensure continuous learning signals even on challenging samples that would otherwise yield zero rewards.

### 2. Comprehensive Hyperparameter Ablations

We conducted extensive ablation studies on:

- Number of curriculum stages (2, 3, 4 stages)
- Hint ratio α (10%, 25%, 40%)
- Accuracy threshold τ (20%, 30%)
- Calibration parameters (sampling number n and temperature T)

Our experiments show that the method is robust to hyperparameter changes. The optimal configuration is 3 stages, α=25%, and τ=20%, with all hyperparameters held constant across models for fair comparison.

### 3. Addressing Guided Prompting Concerns

To address concerns about potential capability weakening, we conducted direct comparisons between models trained with and without guided prompting on completely hint-free test sets. The results strongly refute the dependency hypothesis: models with guided prompting show substantial improvements across all benchmarks (1.5B model: +12.1% average; 7B model: +4.0% average), with particularly strong gains on challenging benchmarks like AIME24 and Olympiad Bench. This demonstrates that guided prompting serves as effective reasoning scaffolding rather than creating problematic dependencies.

### 4. Generalizability Beyond GRPO

We validated ACS with **PPO** on Qwen2.5-Math-1.5B, demonstrating significant improvements (+7.2% average) and confirming that our stability-preserving mechanisms generalize beyond GRPO to other RL algorithms.

### 5. Calibration Robustness Analysis

We conducted comprehensive robustness studies showing that our difficulty calibration maintains >93% consistency across different sampling parameters (n and temperature). Additionally, we tracked difficulty overlap across training stages, finding that 79-88% of samples remain in their original difficulty levels, with the highest stability in challenging samples (Level 3: 84-92%), validating our approach's consistency.

## Manuscript Revisions Completed

Based on this feedback, we have:

- Expanded the related work section with comprehensive curriculum RL discussion, clearly positioning our sample adaptation contribution
- Included all ablation results and hyperparameter specifications
- Added the ADARFT and PPO comparison results
- Provided explicit discussion of current domain scope limitations and promising future directions (other domains, progressive hint removal, theoretical guarantees)
- Clarified experimental design details to demonstrate proper isolation of component contributions

We believe these revisions significantly strengthen our work and address all major concerns raised by the reviewers. We are grateful for the rigorous review process that has helped us improve the paper substantially.

---

### Meta-Review · Area_Chair_NxGD · 2026-01-07

**Summary:**

The paper proposes so-called Adaptive Curriculum Strategies to stabilize curriculum-based reinforcement learning for LLM mathematical reasoning via difficulty calibration and guided prompting, and presents extensive empirical results on several math benchmarks.

**Reviewer Concerns:**

The reviewers agree that the problem is important, the phenomenon of curriculum-induced collapse is clearly identified, and the pipeline is concrete and empirically effective, but they also raise consistent concerns about limited positioning within existing RL curriculum literature, insufficient isolation of which components actually cause the stability gains, narrow domain scope restricted to math reasoning, and the lack of theoretical or principled justification for stability. In addition, some reviewers note that baseline comparisons and protocol choices may favor the proposed method, and that potential risks such as distribution shift, hint leakage, and robustness of calibration are not fully resolved despite added experiments.

**Reviewer Scores:**

Given that these issues affect the generality, interpretability, and scientific grounding of the contribution, and that the rebuttal does not fully address all of them at a conceptual level, the recommendation is reject.

---

### Decision · Program_Chairs · 2026-01-26

Reject